# Error-driven upregulation of memory representations
Alexander Weuthen [1,2,3,4] ✉, Hans Kirschner [1] & Markus Ullsperger [1,3,4,5]

Learning an association does not always succeed on the first attempt. Previous studies associated increased error signals in posterior medial frontal cortex with improved memory formation. However, the neurophysiological mechanisms that facilitate post-error learning remain poorly understood. To address this gap, participants performed a feedback-based association learning task and a 1-back localizer task. Increased hemodynamic responses in posterior medial frontal cortex were found for internal and external origins of memory error evidence, and during post-error encoding success as quantified by subsequent recall of face-associated memories. A localizer-based machine learning model displayed a network of cognitive control regions, including posterior medial frontal and dorsolateral prefrontal cortices, whose activity was related to face-processing evidence in the fusiform face area. Representation strength was higher during failed recall and increased during encoding when subsequent recall succeeded. These data enhance our understanding of the neurophysiological mechanisms of adaptive learning by linking the need for learning with increased processing of the relevant stimulus category.

Forming memories and using acquired knowledge when needed is an essential cognitive capability. Imagine, for example, a teacher, who is trying to learn the names of students of a new class. For some students, the teacher will remember the names right away, but for others the teacher needs several attempts. In this study, we aim to better understand how the brain monitors learning failures and facilitates subsequent association memory formation.

Based on the assumption that successful memory recall requires successful memory encoding, previous neuroimaging studies have investigated the subsequent memory effect by determining which neurophysiological signals at time of encoding predict later recall success[1–3]. Cognitive processes and brain regions contributing to the subsequent memory effect have been differentiated into content-processing regions in the fusiform gyrus (FG) and left inferior frontal gyrus (IFG), attention during encoding in premotor cortex (PMC) and posterior parietal cortex (PPC), as well as storage function in medial temporal lobe regions such as hippocampus and amygdala[4]. There is, however, a lack of studies investigating how the brain monitors failed learning attempts and implements necessary adjustments, such as increased attention and brain network states for improved memory formation. Based on the broader literature on performance monitoring in speeded choice reaction time tasks, the posterior medial frontal cortex (pMFC) has consistently been implicated in accumulating evidence of task demands and a respective signaling function indicating the need for

adjustments[5,6]. For example, the magnitude of error-related functional magnetic resonance imaging (fMRI) in pMFC and frontocentral electro-encephalography (EEG) signals was shown to be predictive for successful performance adaptations[7,8]. Interestingly, the function of pMFC in detecting memory demands and enhancing attention has been overlooked in previous studies, although hemodynamic responses in pMFC were also found to be increased during successful learning in above meta-analysis on the subsequent memory effect[4]. While error-related signals in this region have been associated with improved associative learning[9,10], there is a lack of studies investigating how brain regions involved in performance monitoring and memory formation interact.

The pMFC region can be parcellated into more fine-grained subregions such as anterior and posterior midcingulate, (pre-)supplementary motor and dorsomedial prefrontal cortices based on criteria such as to cytoarchitectonic profiles[11,12]. These regions have been assigned to contribute to large-scale brain networks, such as a midcingulo-insular salience/ ventral attention network, a lateral frontoparietal/ executive control network and a medial frontoparietal default mode network[13,14]. The ventral attention network has been proposed to switch between the frontoparietal control network for external attention and upregulated default mode network for internal attention[15]. However, precise mapping of functional representations in the pMFC onto its anatomical subregions has proven difficult in

[1]Institute of Psychology, Otto-von-Guericke-University Magdeburg, Magdeburg, Germany. [2]Department of Psychiatry and Psychotherapy, Jena University Hospital/Friedrich-Schiller-University, Jena, Germany. [3]German Center for Mental Health (DZPG), partner site Halle-Jena-Magdeburg, Germany. [4]Center for Intervention and Research on adaptive and maladaptive brain Circuits underlying mental health (C-I-R-C), Halle-Jena-Magdeburg, Germany. [5]Center for Behavioral Brain Sciences, Magdeburg, Germany. ✉e-mail: Alexander.Weuthen@uni-jena.de

human fMRI research. This is partly driven by substantial interindividual variability of pMFC anatomy[16]. Therefore, we refer to the pMFC as a broad region related to performance monitoring processes, which may be assigned to different large-scale brain networks[14,17].

While previous studies have speculated on the mechanisms for the attentional allocation of error-driven learning improvements[18], the current study modelled the level of evidence for processing the memory-relevant cue category. Improved external attention should support the extraction of to-be-learned stimulus features and increased internal attention following stimulus presentation should strengthen perceptual representations via mental rehearsal. During these memory formation epochs, neurophysiological processing of the memorized stimulus category should be increased when more attention is allocated on a stimulus. The detection of a memory error should lead to increased stimulus processing and a higher likelihood that the presented association will be remembered. Multivariate decoding may be a useful tool to capture the strength of and evidence for stimulus representations during different phases of memory formation. Previous studies have shown that the degree of behavioral relevance of a presented stimulus category can be decoded during respective cognitive tasks[19,20] and that stimulus decodability is related to the degree how much attention is allocated[21]. While most task-based fMRI studies have used multivariate pattern analyses to compare decoding accuracies for a set of stimuli, it has been suggested that the decision function of multivariate models contains a more fine-grained pattern of stimulus evidence[22], which can be used to determine single-trial differences in stimulus processing and decodability. Here, we tested the hypothesis that regions associated with the monitoring of memory performance, such as pMFC, reflect upregulated selective attention, as approximated by single-trial evidence of stimulus-processing in stimulus-specific regions. The current study investigates face-processing evidence in the ventral visual stream, because of its well-described topography in the posterior and mid-fusiform gyrus[23,24], often referred to as fusiform face area (FFA)[25]. If pMFC links memory-related demand detection, upregulated FFA-based face-processing and improved memory success, this will

improve the understanding how brain networks for performance monitoring, stimulus-based attention and memory formation interact.

## Methods

### Participants

30 young adults (15 male and 15 female participants according to self-reported sex, age 18–35 years, no data on race/ethnicity collected) participated in the current fMRI study after checking inclusion criteria (body mass index between 20 and 30 kg/m², non-smokers, no history of psychiatric or neurological disorders, no metal implants) via phone interview. Participants gave written informed consent before the study began and were compensated with study credits or money (10 EUR per hour) for their time. They obtained written instructions on the behavioral tasks and task comprehension was checked within a practice session outside the scanner. Next, participants were positioned in the MRI scanner. The keyboard was placed under the right hand, a photoplethysmography sensor on the left middle finger and a breathing belt around the chest on the position of the highest elevation. The study was approved by the ethics committee of the medical faculty at Otto-von-Guericke University Magdeburg, Germany. The study protocol and analyses were not preregistered.

### Stimuli

Publicly available images of emotionally neutral faces from the Picture Database of Morphed Faces[26] and house images from the DalHouses sample[27] were used. The background color of the house images was replaced with the same grey scale as in the face images. The tasks also contained eight differently tilted gabor patch stimuli with an orientation point in the extension of the middle white stripe rendered with Psychtoolbox 3 with Matlab 2018a on a Windows 10 computer.

### Behavioral tasks

In the feedback-based association learning task (see Fig. 1a), the 30 participants learned to associate faces with gabor patches in eight possible

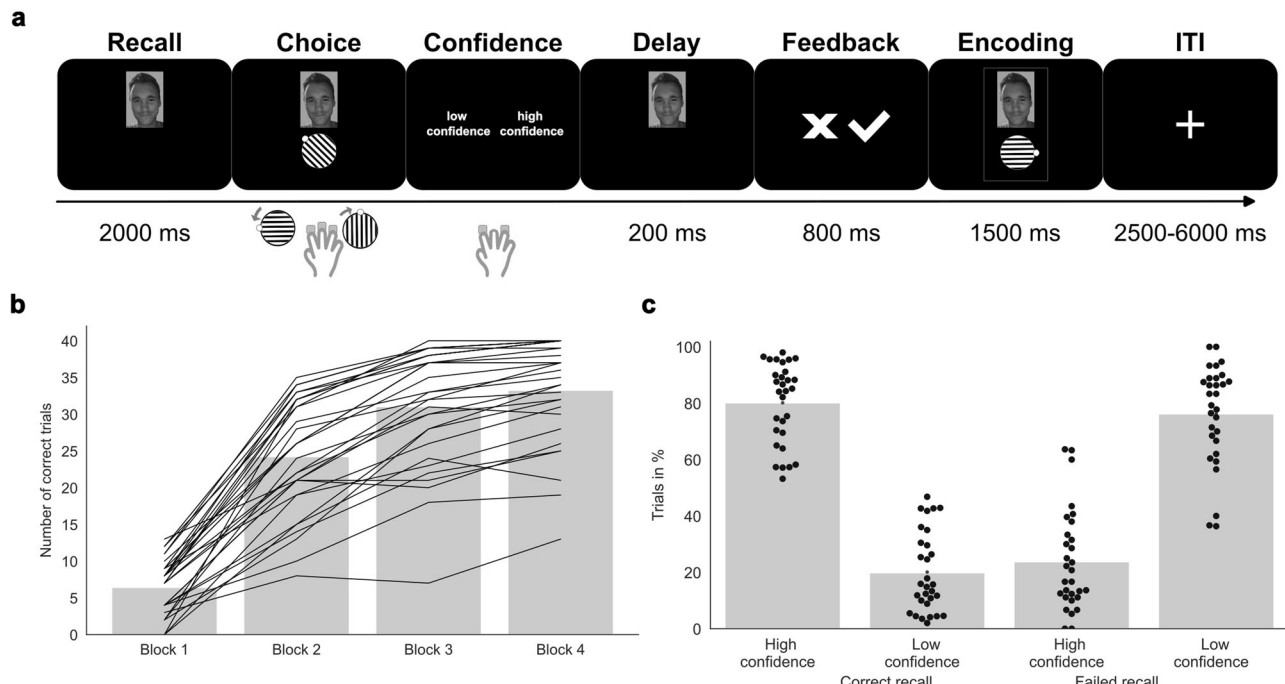

**Fig. 1 | Trial structure and behavioral results of the feedback-based association learning task. a** In a continuous learning experiment, 30 participants (30 male, 15 female) learned to associate faces and eight different orientations of gabor patches. During a trial, participants chose the presumed orientation, selected a low or high level of confidence and obtained either positive or negative feedback. Finally, the correct combination of face and gabor patch was presented as a learning opportunity

for trials showing the same face in later blocks, followed by a jittered inter-trial-interval. **b** Participants successfully learned the presented associations and recalled the matching orientation of the gabor patch better in later repetitions of a face. **c** In most of the trials, participants were able to distinguish successful and failed memory recall, indicating reasonable meta-memory performance.

orientations. Each trial began with an inter-trial-interval showing a fixation cross in the middle of the screen for a jittered duration between 2500 and 6000 milliseconds. Then, a face stimulus was presented and 1000 milliseconds later a gabor patch appeared in a random but incorrect orientation. Participants had to choose the matching orientation with their right index finger for a left-directed rotation and right ring finger for a right-directed rotation. If they saw a face for the first time, they were instructed to make a guess. On subsequent encounters of the face, they should recall the associated orientation from their memory. After confirming their choice with the right middle finger, low and high confidence options were presented on screen, such that participants could indicate their recall certainty with respective index and ring finger presses. The side of the presentation for low-confidence and high-confidence ratings was altered for each trial. After 200 milliseconds delay, based on recall success, either positive or negative feedback was presented for 800 milliseconds. At the end of each trial, the correct combination of face and gabor patch was presented for 1500 milliseconds for (re-) encoding. Each face was presented four times, with at least two and a maximum of 15 trials until the next trial with the same face. The task consisted of five independent runs with eight new faces each, summing up to 160 trials in total. Between runs, participants were presented with a pause screen on which the relative number of correct trials was displayed. The next run with eight new face stimuli was resumed with a confirmation button press.

In the 1-back localizer task, on each trial, the 30 participants were presented a face or a house together with a gabor patch in one of eight possible orientations. They were instructed to attend and compare both stimuli with the stimulus combination shown in the directly preceding trial, and to press the confirmation key as fast as possible when the presented stimulus combination was a direct repetition. Presentation times were analog to the durations of encoding with 1500 milliseconds and the inter-trial-interval jittered between 2500 and 6000 milliseconds as in the feedback-based association learning task. Within each run, two new face and two new house stimuli were presented four times each, summing up to 80 trials for five runs in total. Direct repetitions occurred in two of 16 trials per run to keep participants engaged with attending, encoding and rehearsing the presented stimuli.

## Data acquisition
Magnetic resonance imaging (MRI) data were obtained by a 3 Tesla Siemens Prisma scanner with a 64-channel head coil. After brief anatomical scout images, structural MRI data were assessed using a magnetization-prepared rapid gradient echo sequence in sagittal slices (voxel size = 1 × 1 × 1 mm, matrix size = 192 × 256 × 256, repetition time = 2.5 s, echo time = 0.00282 s, flip angle = 7°, multiband factor = 2). While participants performed the feedback-based association learning task and the localizer task, fMRI scans were recorded with a field of view aligned to anterior and posterior commissures (voxel size = 2.2 × 2.2 × 2.2 mm, matrix size = 100 × 100 × 66, repetition time = 2.0 s, echo time = 0.03 s, flip angle = 80°, multiband factor = 2, interleaved order, no interslice gap). Single-band reference images were recorded on the first and field maps after the last functional scan. Due to technical issues, one participant lacked the single-band reference image and two participants lacked peripheral physiological recordings.

## fMRI preprocessing
MRI data were converted using dcm2niix (version v1.0.20190902), and renamed in accordance with Brain-Imaging-Data-Structure format[28]. Data were analyzed on a high-performance computing cluster using Linux Debian (version 4.9.0-16-amd64). For preprocessing, fMRIPrep version 23.2.2[29] was run with a singularity image (version 2.6.1-dist) wrapped around a docker container. Preprocessing encompassed slice time correction, susceptibility distortion correction, boundary-based registration and spatial normalization to obtain images in MNI152NLin2009cAsym output space, keeping the size of 2.2 mm³ voxels. Further details on the fMRIPrep-based preprocessing pipeline can be found in the section Supplementary Methods. Physiological regressors for retrospective image correction of

respiratory and cardiac confounds were obtained from the PhysIO package in the TAPAS toolbox[30]. For simultaneous denoising and fitting of event-related hemodynamic response functions, general linear models on the preprocessed images contained following confound regressors: 24 motion parameters (six rigid body motion parameters, six derivatives, and respective twelve squared motion parameters), 18 physiological regressors (six cardiac, eight respiration, four combined cardiac and respiration), ten anatomical component correction regressors (five white matter, five cerebrospinal fluid), the global signal, a cosine drift model and a constant intercept.

## Analysis software
Behavioral and fMRI analyses were based on custom Python (version 3.8.12) code within Jupyter Lab (version 3.4.8), using numerical processing and statistical testing with Numpy (version 1.23.5), Scipy (version 1.10.0), Pandas (version 1.5.3) and Statsmodels (version 0.13.2), plotting functions from Matplotlib (version 3.6.3) and Seaborn (version 0.12.0), and decoding tools from Scikitlearn (version 1.2.1) and Nilearn (version 0.10.0[31]). Visualization of fMRI results was based on MRIcroGL (version 1.2.20220720b).

## Behavioral analyses
In the feedback-based association learning task, for in total 160 trials in four blocks and because participants had to guess in the first block, there were maximally 120 trials in which participants could remember the correct orientation of the associated gabor patch from a past learning opportunity. A one-sample $t$-test against a chance level of 12.5% was performed for the relative number of correct trials per participant, to determine whether the presented face and gabor patch associations were successfully learned. The performance increase between different blocks was assessed with a one-way analysis of variance and post-hoc tests with Tukey's honestly significant differences. Participant's meta-memory performance ($d_{Prime}$) was assessed as the average of the probability distribution between the proportion of high-confidence selections upon successful recall (sensitivity) and the proportion of low-confidence selections in failed recall trials (specificity). Sensitivity and specificity probability distributions functions were adjusted for infinite values by subtracting the proportion of one correct or incorrect trial, respectively. Meta-memory performance $d_{Prime}$ and bias $d_{Bias}$ were tested for significance with a one-sample $t$-test.

In the 1-back localizer task, there were ten repetition trials on which participants had to press the confirmation key and 70 non-repetition trials where they were instructed to attend and encode the presented stimuli but not to press. According to signal detection theory, trial types were distinguished into hits for a correct press on a repetition, misses for a non-press on a repetition, correct rejections for a non-press on a non-repetition and false alarms for a press on a non-repetition. Task performance was evaluated based on hit and correct rejection rates and significance was tested using one-sample $t$-tests. Insufficient task comprehension of a participant was assumed for outliers, which were defined by a task performance being two standard deviations ($SD$) lower than the average ($M$) performance of all participants.

## fMRI analyses
In the feedback-based association learning task, univariate general linear model fMRI analyses were conducted by simultaneously fitting a hemodynamic response function using the Glover model convolved with respective event regressors during memory recall ($Error_{ConfidenceLow}$ or $Error_{ConfidenceHigh}$ or $Correct_{ConfidenceLow}$ or $Correct_{ConfidenceHigh}$), confidence selection (low or high), feedback presentation (positive or negative), encoding as determined by current and subsequent recall success (Error-Error or ErrorCorrect or CorrectCorrect or CorrectError in combination with respective confidence levels). Two general linear model analyses were performed, one for the post-error subsequent memory effect and one for memory-error detection.

In the first general linear model, neurophysiological signals related to recall, confidence and feedback were included and each trial type was

convolved as a separate regressor, such that the shared variance is encompassed in the residual variance of the model. Multicollinearity between convolved regressors was examined using the variance-inflation-factor index, assuming moderate multicollinearity for values > 5 and < 10, and high multicollinearity for a variance-inflation-factor > 10, and using Pearson correlation values below < 0.90 following current practices to indicate sufficiently efficient design matrices[32,33]. To determine the brain regions associated with performance monitoring of memory errors, fMRI contrasts were calculated for hemodynamic responses upon failed recall ($Error_{LowConfidence} > Correct_{HighConfidence}$) as implicit indication for a detected demand of better memory formation, the selection of recall uncertainty (low > high confidence) as discrete internal memory error evidence, and the presentation of memory error feedback (negative > positive) as external evidence.

In the second general linear model, encoding regressors were used together with regressors for recall and for confidence while the feedback-related regressors were excluded because of the redundancy and temporal overlap with encoding regressors. To ensure that participants were aware of required memory demands before successful re-learning, the post-error subsequent memory effect was calculated between low-confident error trials which were later remembered with a high level of confidence and those error trials with subsequent failed recall and low confidence ($Error_{LowConfidence}Correct_{HighConfidence} > Error_{LowConfidence}Error_{LowConfidence}$). To investigate neurophysiological associations of reconsolidation processes and error-driven learning successes, an additional analysis assessed differences between subsequent repeated correct and initial correct trials ($Correct_{HighConfidence}Correct_{HighConfidence} > Error_{LowConfidence}Correct_{HighConfidence}$).

In the 1-back localizer task, the univariate general linear model analysis consisted of hemodynamic response functions convolved for faces and houses which were further differentiated into eight regressors based on four different signal detection theory trial types (hit, miss, correct rejection, false alarm), and denoising parameters as described in the section fMRI pre-processing. To determine which brain regions are systematically related to face-processing, a contrast on correct non-press trials ($Face_{CorrectRejection} > House_{CorrectRejection}$) was calculated. The topography of significant clusters in FG was visually compared regarding its overlap with probabilistic cytoarchitectonic maps for right FG-2 and FG-4 regions[34].

Upon statistical testing of the group results in a second-level general linear model, contrast maps were smoothed with an 8 mm kernel and a voxel-wise false-discovery rate threshold was applied, removing clusters with an extent of less than five continuous voxels (equivalent to clusters of at least 53.24 mm³).

To assess in which brain regions memory error monitoring processes converge, a conjunction analysis was performed by identifying voxels with significant effects in all contrasts i.e., failed recall, a low level of confidence and negative feedback. The conjunction effects were statistically tested against a chance level of $p < 0.05$ based on 10000 random permutations[35]. To show overlaps, such as between memory error monitoring-related hemodynamic responses and the post-error subsequent memory effect, the statistical results were overlayed with the conjunction image and plotted together with the contrasts for negative feedback as the largest topographical extent and most explicit evidence level on memory errors.

Related to topographical variations in the cluster locations, associations with an often-used seven-networks brain parcellation scheme[14] were assessed, encompassing visual, somatomotor, dorsal attention, ventral attention, limbic, frontoparietal and default networks in the cerebral cortex. Thresholded statistical maps were overlayed to color-coded brain maps for these networks and the relative number of voxels assigned to a mask was quantified. Voxels falling outside the cerebral network masks were specified as unassigned.

**Multivariate cross-classification**

A key aim of the current study was to develop a quantitative proxy measure for stimulus-based attention as a link between error-driven demand detection and encoding success. For each participant, a multivariate model on face-processing was trained in the localizer task and later applied to memory-related epochs in the feedback-based association learning task, such as memory recall, encoding and potential rehearsal processes in the inter-trial-interval. First, the univariate general linear models described in the previous sections were adapted for single-trial deconvolution according to the least-squares separate approach[36] to obtain a series of beta-maps. In this regard, all correct rejection face and house trials in the localizer task were determined and stimulus presentation of each trial was once defined as target event in an additional, independent general linear model. The target trial was convolved with a hemodynamic response function as a separate regressor, while controlling for all other events and denoising parameters such as in univariate general linear model analyses. In case participants showed optimal performance in the localizer task (i.e., they correctly identified all repetitions and did not display false alarms) a total of 70 single-trial ($M = 69.13$, $SD = 2.29$) beta-maps could be derived.

Based on the univariate fMRI results in the localizer task, bilateral cytoarchitectonic probability masks for FG-4 showed a strong overlap with increased hemodynamic responses during face processing. After smoothing with a 6 mm full-width at half-maximum kernel, deconvolved betaseries of voxels within the FG-4 mask were extracted and trials were labeled for five folds according to the presented run in the task. A balanced, probability-scaled linear support vector machine ($C = 1$) with a squared penalty function was trained on four of the five runs to predict whether trials from the left-out run were either faces or houses. Within the five-fold leave-one-run-out cross-validation, a standard scaler ($M = 0$, $SD = 1$) was fit to the four training runs and applied to the left-out run. Univariate feature selection was applied by maintaining only the beta-weights of the 14 voxels with the strongest positive analysis of variance effects, to obtain results for participant-specific FFA voxels and to reach a feature-to-sample ratio of approximately 1:5 before fitting the support vector machine. Feature selection was only based on the training samples, both during cross-validation and cross-classification, to prevent leakage and overfitting. Decoding accuracies were evaluated by testing whether the average accuracies of the five runs per participant exceeded a chance level of 50% with a one-sample $t$-test. Face and house trials were tested for equal decoding accuracies with a $t$-test for dependent samples to ensure that the FFA-based face-processing model was balanced and did not prefer either of the two categories.

After leave-one-run-out cross-validated model evaluation, trials from all five folds were included in model training. A full model was fit on correct rejection trials of all localizer task runs of a participant with the same scaling procedure and feature selection as during cross-validation. The support vector machine was fit on the 14 selected voxels of up to 70 correct rejection trials of all localizer runs of a participant, to be applied to the memory epochs for trials in the feedback-based association learning task. Single-trial deconvolution and selection of FFA voxels was repeated for the 160 trials in the feedback-based association learning task and the three memory-relevant epochs of stimulus recall, encoding and inter-trial-interval. To ensure sufficient independence and additional variance explanation of the epochs, a step-wise procedure was chosen, such that in the recall-related deconvolution only recall events, in the encoding-related deconvolution both recall and encoding events, and in the inter-trial-interval-related deconvolution all three events were included.

The machine learning model for evaluating the strength of stimulus representations then predicted the presented class and estimated the probability of face-processing for each trial in each of the epochs. To evaluate the validity of the single-trial face-processing model, the correspondence between the average probability of the face-class and the absolute number of predicted face-class trials was controlled by significance tests for the Spearman correlation coefficient. To assess which other regions are potentially involved in allocating attention to the presented stimulus category, the classifiers single-trial decoding probability of FFA-based face-

processing evidence was then fit to all other voxels in a whole-brain general linear model analysis on the single-trial betaseries for each of the memory-relevant epochs (recall, encoding, inter-trial-interval), respectively. The predicted face class-probability parameter was compared between different behaviorally assessed trial types during stimulus recall, encoding and inter-trial-interval, to determine whether the proxy measure for allocated stimulus-based attention is increased after memory errors and related to a higher likelihood of successful memory formation, as determined by later recall success. For each of the three memory epochs, in a linear mixed model the representation strength measure was fit to the regressors encoding demand and subsequent recall success, while restricting the analysis to ErrorError, ErrorCorrect and CorrectCorrect trials, and controlling within-participant dependencies by using participant as group factor. In particular, the encoding demand regressor was set to a value of 0 for CorrectCorrect trials, and to a value of 1 for ErrorError and ErrorCorrect trials. The regressor for subsequent recall success was set to a value of 0 for ErrorError trials, and to a value of 1 for ErrorCorrect and CorrectCorrect trials. The face-class processing measure was based on a Platt-scaled probability measure of the linear support vector machine, and was able to range between 0 and 1, where 1 indicates the highest level of multivariate evidence for FFA-based face-processing. In this way, beta-weights from the linear mixed model corresponded to values interpretable as increased face-processing evidence according to a change in encoding demand and in association with subsequent recall success.

### Reporting summary
Further information on research design is available in the Nature Portfolio Reporting Summary linked to this article.

## Results
### Performance on the feedback-based association learning task
During continuous fMRI scanning, participants ($n = 30$) performed a feedback-based association learning task, in which they had to learn which of eight orientations of a gabor patch is associated to a set of unknown faces. Each trial consisted of a recall phase where only the face was presented, the choice of a presumed orientation, the selection of a low or high recall confidence, the presentation of performance-based feedback, and the display of the correct combination of the face and the associated gabor patch, followed by an inter-trial-interval, which may have offered a chance for stimulus rehearsal (Fig. 1a). Each participant performed five independent runs, in which eight new faces were learned and repeated in three more blocks. Across all runs and blocks in the feedback-based association learning task, participants correctly remembered the associations between faces and the eight different orientations of the gabor patches in 59.35% of trials [$t(29) = 16.88$, $p < 0.001$, one-sample $t$-test > 12.5% chance level, $R^2 = 0.905$, $CI_{95} = 54.63$ to $64.07\%$)]. Memory performance was improved for face repetitions in later blocks [$F(3,26) = 93.53$, $p < 0.001$, one-way analysis of variance, $R^2 = 0.702$; for further statistics on pairwise comparisons and confidence intervals, see Supplementary Table 1]. Correct trials were associated with ratings of high confidence on 80.14% ($SD = 14.28$) of trials, and for failed recall participants selected a low level of confidence on 76.22% ($SD = 17.78$) of trials (Fig. 1c). Participants had good meta-memory performance [$d_{Prime} = 0.95$; $t(29) = 11.76$, $p < 0.001$, one-sample $t$-test > 0, $R^2 = 0.822$, $CI_{95} = 0.82$ to $1.09$], without an indication of a bias towards under- or overconfidence [$d_{Bias} = -0.03$; $t(29) = 0.24$, $p = 0.811$, one-sample $t$-test, $R^2 = 0.002$, $CI_{95} = -0.02$ to $-0.03$]. Together, these results suggest that participants learned to associate faces with tilted gabor patches and gained accurate confidence levels in a feedback-based association learning task.

### Implicit and explicit evidence for memory errors is represented in pMFC
In the feedback-based association learning task, on each trial participants accumulated evidence on the quality of a current memory representation,

both implicitly and explicitly. First, they attempted to recall the correct association to a face. Thereafter, they indicated their confidence in their response (binary variable low vs. high confidence). Finally, they received feedback regarding the correctness of the chosen orientation of the gabor patch. Accordingly, there were three different epochs for modelling neurophysiological correlates on the monitoring of memory errors. Univariate general linear model analyses showed increased hemodynamic responses in pMFC at all stages of error monitoring in the task (Fig. 2). These epochs showed pMFC effects during failed recall [$Error_{ConfidenceLow} > Correct_{ConfidenceHigh}$; $z(29) = 4.30$, $p_{FDR} < 0.001$, $R^2 = 0.389$, $CI_{95} = 3.94$ to $4.66$; x = 5, y = 22, z = 40, Supplementary Table 2], the selection of recall uncertainty [low > high confidence; $z(29) = 4.87$, $p_{FDR} < 0.001$, $R^2 = 0.450$, $CI_{95} = 4.51$ to $5.23$; x = 3, y = 35, z = 36, Supplementary Table 3] and the presentation of memory-error feedback [negative > positive feedback; $z(29) = 6.80$, $p_{FDR} < 0.001$, $R^2 = 0.615$, $CI_{95} = 6.44$ to $7.16$; x = 5, y = 17, z = 51, Supplementary Table 4]. Variance-inflation-factor indices were < 5 and $R_{Pearson} < 0.90$ between the convolved design matrix regressors, indicating sufficiently low multicollinearity in the univariate general linear model analysis (Supplementary Fig. 1). A conjunction analysis on all three error monitoring contrasts showed overlaps in a pMFC region encompassing both hemispheres, bilateral dorsolateral prefrontal cortices (dlPFC) and right premotor cortex (PMC). Most of the significant voxels ( > 90%) in the error monitoring conjunction were assigned to the frontoparietal control network. The overlap, in the cluster location of regions such as pMFC in all three epochs, suggested that pMFC's presumed function in performance monitoring also applies to tracking internal and external evidence of currently inaccurate memory representations.

### Error-related pMFC activity predicts successful subsequent recall
After an attempted recall in the feedback-based association learning task, participants had another learning opportunity, in which the correct association of the presented face and gabor patch was displayed. For failed recall trials, a univariate general linear model analysis determined which neurophysiological differences during post-error encoding epochs distinguish successful and failed subsequent recall. Results for the post-error subsequent memory effect showed increased hemodynamic responses in pMFC [$Error_{LowConfidence}Correct_{HighConfidence} > Error_{LowConfidence}Error_{LowConfidence}$; $z(29) = 5.37$, $p_{FDR} < 0.001$, $R^2 = 0.499$, $CI_{95} = 5.01$ to $5.73$; x = -3, y = 0, z = 71], and replicated regions previously reported for the subsequent memory effect, such as IFG, FG, PPC and PMC (Fig. 3 and Supplementary Table 5). An involvement of the hippocampus was found in an additional analysis on memory reconsolidation [$Correct_{HighConfidence}Correct_{HighConfidence} > Error_{LowConfidence}Correct_{HighConfidence}$, see Supplementary Fig. 2 and Supplementary Table 6]. The location of the post-error subsequent memory effect in pMFC was posterior and superior of the conjunction for memory error monitoring, where it overlapped with contrast for negative compared to positive feedback. Regarding network correspondences, the post-error subsequent memory effect showed a more variable pattern compared to the error monitoring conjunction contrast. More than 5% of significant voxels were assigned to ventral attention, dorsal attention, visual, default and frontoparietal networks (see Fig. 3d). While the function of pMFC for memory formation has been neglected in a previous meta-analysis[4], the overlap with memory-error related regions suggested a preparatory role for an adaptive learning state. Successful post-error learning improvement has been related to increased error-related fMRI and EEG signals before. Yet, the correlational nature of these results precludes a better understanding of the underlying mechanisms. We suggest that a candidate mechanism is increased processing of information relevant to resolve the problem at hand. To test this idea, we applied a model on the strength of stimulus representations as a marker of allocated attention to the presented stimulus category.

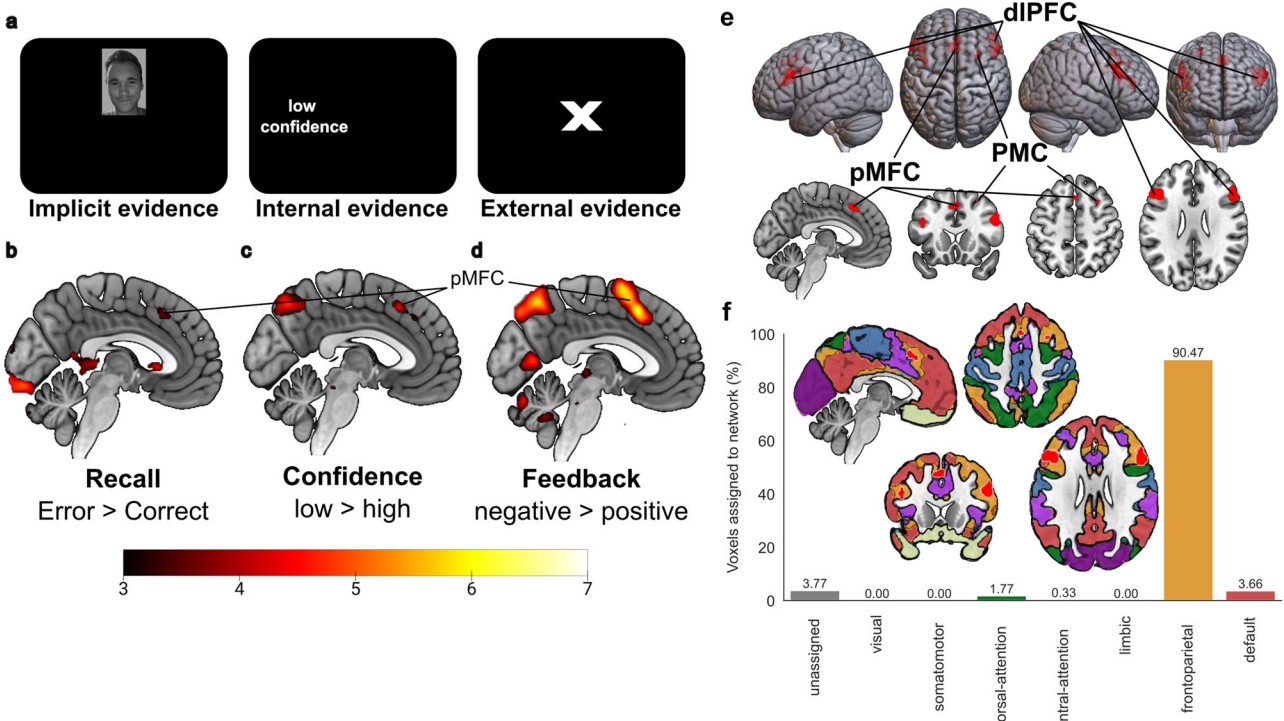

**Fig. 2 | Hemodynamic responses related to the monitoring of memory errors in the feedback-based association learning task. a** All three types of memory-error evidence, such as **b**, failed recall attempts, **c** the selection of low confidence and **d** negative feedback showed increased hemodynamic responses in posterior medial frontal cortex (pMFC) in a univariate general linear model analysis of the 30 participants (15 male, 15 female). The color bar indicates z-scores and the results are thresholded using a false discovery rate ($q < 0.05$). **e** The conjunction image shows voxels which had significant hemodynamic responses in all three memory error monitoring contrasts (failed recall, low confidence, negative feedback). The effects converged in pMFC, right premotor cortex (PMC) and bilateral dorsolateral prefrontal cortex (dlPFC). **f** Network correspondence was quantified according to the overlap of voxels in the conjunction with a seven-networks cortical parcellation scheme[14]. Most voxels of the error monitoring conjunction were assigned to the lateral frontoparietal control network.

## The 1-back localizer task captured face-selective processing in a cytoarchitectonic mask of the fusiform gyrus

To build a model of stimulus representation strength, we trained a classifier to distinguish fMRI data during face and house stimulus presentations in a 1-back localizer task (Fig. 4a). 28 out of 30 participants performed the task either without mistakes or within two standard deviations from the group average (Fig. 4b). Univariate general linear model analyses in the 1-back localizer task showed that hemodynamic responses were larger for faces than houses in FFA, as determined by a strong overlap with cytoarchitectonic probability maps of left and right FG-4 [$Face_{CorrectRejection}$ > $House_{CorrectRejection}$; $z(27) = 5.48$, $p_{FDR} < 0.001$, $R^2 = 0.527$, $CI_{95} = 5.11$ to 5.85; $x = 44$, $y = -46$, $z = -27$], but also in other regions previously described as face-selective such as superior temporal sulcus [$z(27) = 4.48$, $p_{FDR} < 0.001$, $R^2 = 0.426$, $CI_{95} = 4.11$ to 4.85; $x = 51$, $y = -46$, $z = 5$] and anterior temporal lobe [$z(27) = 4.68$, $p_{FDR} < 0.001$, $R^2 = 0.448$, $CI_{95} = 4.31$ to 5.05; $x = 40$, $y = 19$, $z = -31$, Fig. 4 and Supplementary Table 7]. Increased hemodynamic responses for houses compared to faces were found in regions among parahippocampal gyrus [$House_{CorrectRejection}$ > $Face_{CorrectRejection}$; $z(27) = 7.02$, $p_{FDR} < 0.001$, $R^2 = 0.646$, $CI_{95} = 6.65$ to 7.39; $x = -29$, $y = -52$, $z = -5$; Fig. 4 and Supplementary Table 8]. Overall, univariate fMRI results in the localizer task displayed the classical dissociation in the ventral visual stream, displaying FFA-related hemodynamic responses being larger for faces and house-specific hemodynamic responses in the parahippocampal gyrus.

In the localizer task, a machine learning model was trained, in order to predict the strength of FFA-based face-processing evidence during memory-relevant epochs in the feedback-based association learning task. Leave-one-run-out cross-validation reached an average balanced decoding accuracy of 73.15% [$t(27) = 12.38$, $p < 0.001$, one-sample $t$-test, > 50%

chance level, $R^2 = 0.855$, $CI_{95} = 69.96$ to 76.33%] on distinguishing faces and houses based on the selection of the 14 voxels within FFA as features, which show the strongest analysis of variance effects. The prediction of house and face stimuli showed no statistically significant difference in the likelihood of the face representation strength models to prefer either of both categories [$t(27) = -0.18$, $p = 0.855$, two-sample $t$-test, $R^2 = 0.001$, $CI_{95} = -0.002$ to $-0.003$]. Taken together, the decoding accuracies and control analyses suggested that the multivariate face-processing model was able to evaluate face-processing evidence by distinguishing face and house trials.

## FFA-based face-memory representations are simultaneously upregulated in a network of cognitive control regions

In the next step, the participant-specific face-processing models, which were trained on trials in the localizer task, were applied to the single-trial beta-series of memory-relevant epochs in the feedback-based association learning task, i.e., the presentation of the faces in the recall phase, processing of the correct face-orientation association during encoding, and a potential rehearsal phase in the inter-trial-interval. The classifier predicted the presentation of face in 72.23% (SD = 19.38) of recall betaseries, in 39.53% (SD = 18.58) of encoding betaseries, and in 17.35% (SD = 18.72) of inter-trial-interval betaseries. The classifier's decoding rates were systematically related to the predicted class probability averages of a participant (Supplementary Fig. 3), during recall [$R_{Spearman}(27) = 0.973$, $p < 0.001$, $R^2 = 0.946$, $CI_{95} = 0.941$ to 0.987], encoding [$R_{Spearman}(27) = 0.932$, $p < 0.001$, $R^2 = 0.869$, $CI_{95} = 0.858$ to 0.969] and inter-trial-interval [$R_{Spearman}(27) = 0.938$, $p < 0.001$, $R^2 = 0.879$, $CI_{95} = 0.868$ to 0.971]. This correspondence was expected and suggested that the probability measure contained information relevant to differentiate the representation strength of memory-relevant stimuli on a single-trial level. Higher

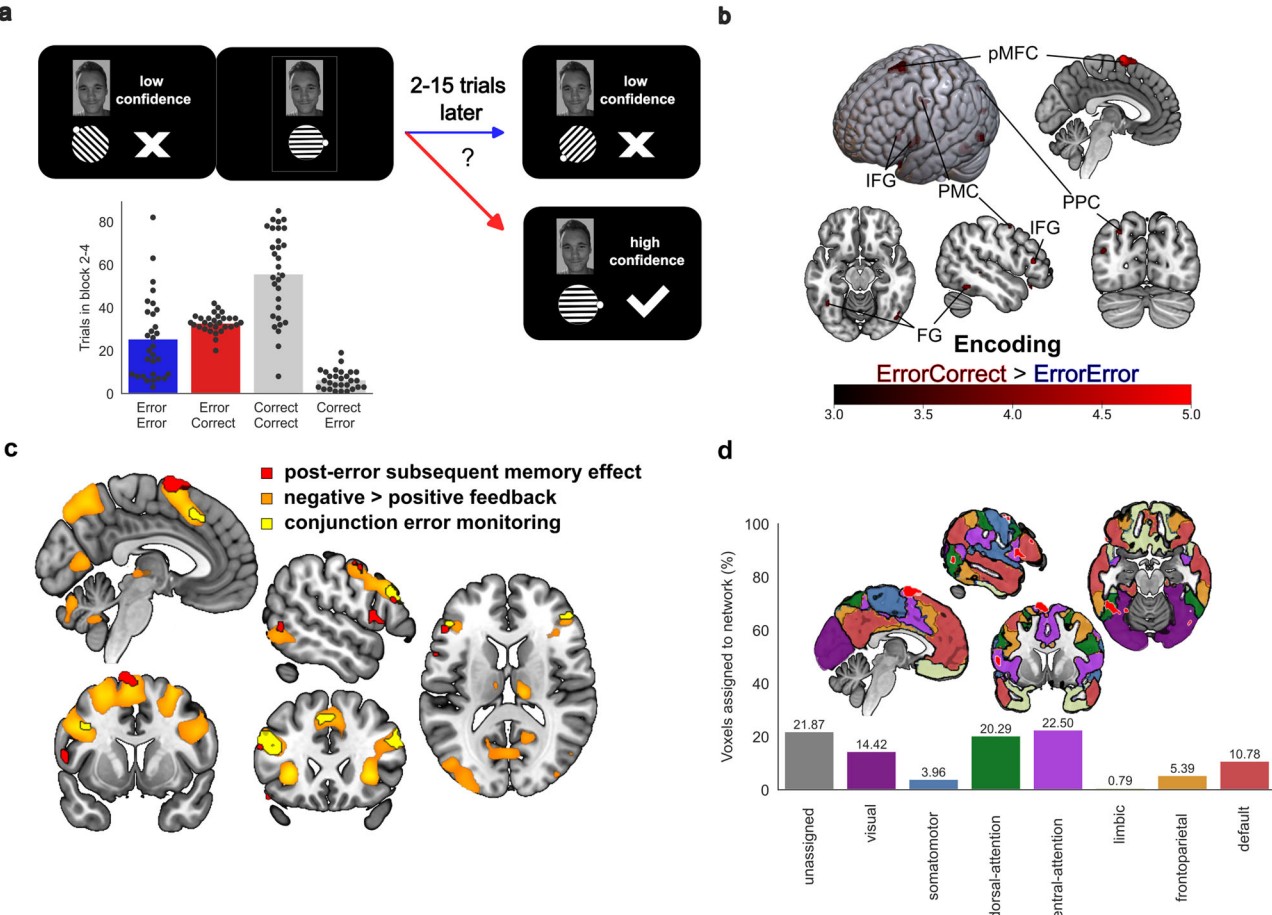

**Fig. 3 | The post-error subsequent memory effect in a univariate functional magnetic resonance imaging (fMRI) analysis. a** The bar plot shows the number of trials per participant ($n = 30$, 15 male, 15 female) for the combination of recall success in the current trial and recall success for the next presentation of the same face. The aim was to distinguish trials with memory (re-)encoding demands which lead to successful memory formation (ErrorCorrect) from failed recall trials which did not lead to successful post-error learning adjustments (ErrorError). **b** Univariate fMRI general linear model results replicated previously described regions from a meta-analysis on the subsequent memory effect[4], showing increased activity during successful recall (Error$_{LowConfidence}$Correct$_{HighConfidence}$) compared to repeatedly failed recall (Error$_{LowConfidence}$Error$_{LowConfidence}$). The color bar indicates z-scores and the results are thresholded using a false discovery rate ($q < 0.05$). **c** Overlap of the post-error subsequent memory effect (red) with the conjunction of error monitoring contrasts (yellow) and the contrast for negative feedback (orange). The pMFC cluster for the post-error subsequent memory effect overlapped with a posterior portion of the cluster related to processing negative feedback, suggesting that its demand-dependent upregulation may have a preparatory function. **d** The image shows the post-error subsequent memory effect plotted as overlay to a seven-networks cortical parcellation scheme[14]. Most voxels were assigned to the ventral attention network, dorsal attention network and visual network.

representation strength for face processing showed the strongest FFA-related hemodynamic responses in a region overlapping with the cytoarchitectonic mask for FG-4 as shown by separate general linear models for recall (Supplementary Table 9), encoding (Supplementary Table 10) and inter-trial-interval (Supplementary Table 11) in the feedback-based association learning task. Higher face-representation strength was also reliably related to regions among pMFC, dlPFC and visual cortex for all three memory-relevant epochs (Fig. 5a). During recall and encoding, bilateral anterior insula showed increased hemodynamic responses related to face-processing evidence. During encoding and inter-trial-interval, associations of single-trial representation strength were also overlapping with bilateral cytoarchitectonic masks of the basal forebrain and the border zone between amygdala and the nucleus basalis of Meynert. The pMFC topography related to face-processing evidence during recall and encoding overlapped with the error-monitoring conjunction (Fig. 6a). A posterior and superior pMFC cluster was found during encoding and the inter-trial-interval. A similar pattern was apparent in dlPFC, showing a more posterior dlPFC cluster during encoding and inter-trial-interval. However, both in pMFC and dlPFC the effects related to face-processing evidence were consistently overlapping with the hemodynamic response for negative feedback processing in all three epochs. These overlaps showed a shared topography

between processes for the monitoring of inaccurate memory representations and upregulated stimulus representations. Assessment of large-scale brain network correspondences indicated the strongest association with the visual network, and to a lesser degree with dorsal attention and frontoparietal networks (Fig. 6b). Overall, multivariate cross-classification analyses highlighted brain network nodes simultaneously upregulated with increased face-processing in face-specific regions of the ventral visual stream. This suggests that these regions work in concert to allocate attention in form of upregulated stimulus representations, which could enhance cognitive operations for association learning, such as extracting memory-relevant stimulus features or improving stimulus maintenance.

## FFA-based face-processing evidence is increased after memory errors and predictive of subsequent recall success

The proxy measure for the strength of stimulus representations, as a marker of allocated attention to the presented stimulus category, was also analyzed regarding its correspondence with behavioral necessity and success on learning the presented associations (Fig. 5b). Previous fMRI general linear model analyses indicated increased hemodynamic responses in pMFC both for error-monitoring processes related to encoding demand and subsequent memory performance related to encoding success. A link between the level

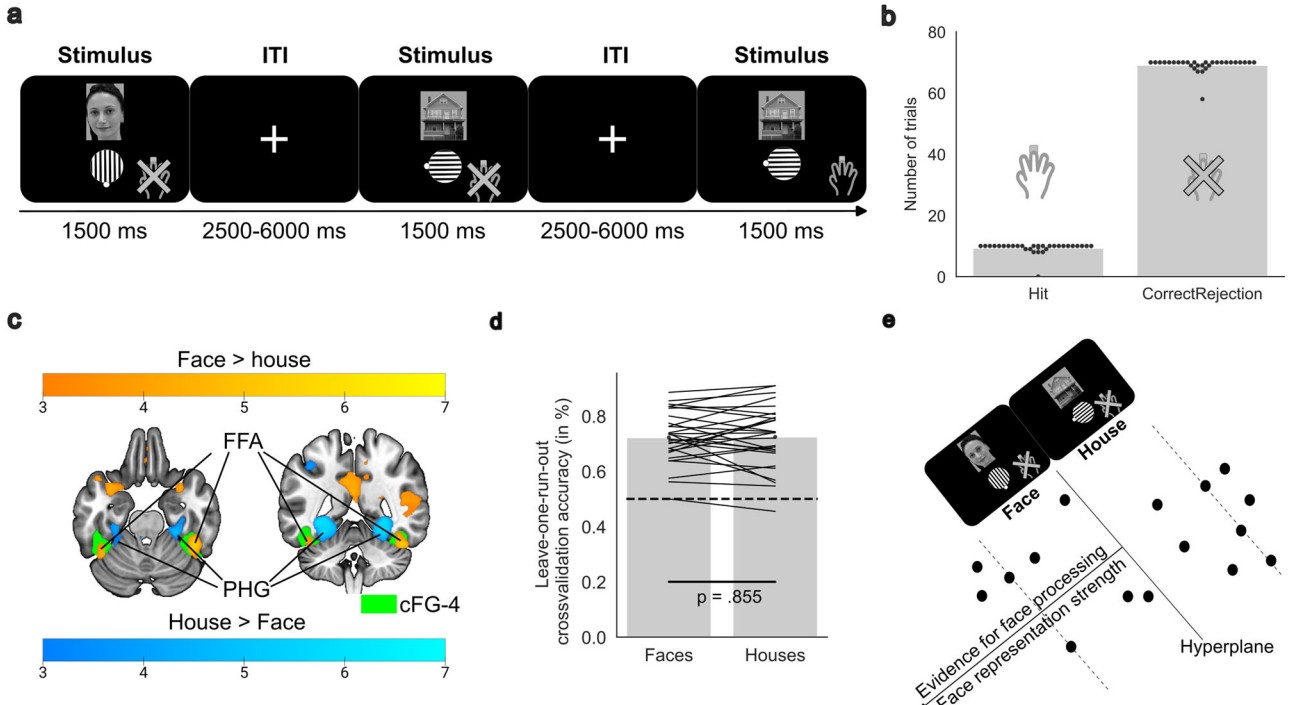

**Fig. 4 | Localizer task trial structure, behavioral results, univariate fMRI analyses and training of machine learning-based face-processing model. a** The 1-back localizer task had comparable presentation times as chosen in the feedback-based association learning task for the stimulus presentation and the inter-trial-interval. In two of 16 trials per run, direct stimulus repetitions occurred. On these repetitions, participants ($n = 30$, 15 male, 15 female) were instructed to quickly press the confirmation key. The task consisted of five runs, each containing two new faces and two new houses. **b** Most participants performed the task without mistakes (misses or false alarms). Two participants were excluded from further fMRI analyses of the localizer task and later multivariate pattern analyses, because task comprehension and attention to the task could not be assured. **c** Conventional general linear model analyses showed that hemodynamic responses were larger for faces than houses in

the right fusiform gyrus, and larger for houses than faces in the parahippocampal gyrus (PHG). The fusiform gyrus cluster largely overlapped with a cytoarchitectonic probability mask for fusiform gyrus 4 (cFG-4) and was identified as fusiform face area (FFA). The color bar indicates z-scores and the results are thresholded using a false discovery rate (q < 0.05). **d** A probability-scaled linear support vector machine was trained to distinguish faces and house based on the 14 strongest voxels in the cFG-4 mask according to analysis of variance feature selection, to quantify FFA-based face-processing evidence. Average classification accuracies during leave-one-run-out cross-validation showed no statistical difference in the prediction for faces and houses. **e** The assessed distance from the multivariate hyperplane indicates evidence for face-processing as shown in the schematic overview.

of stimulus representations strength and both, encoding necessity and success, remained to be assessed. The analyses were restricted to ErrorError, ErrorCorrect and CorrectCorrect trials to determine encoding demand as contrast between current insufficient (ErrorError, ErrorCorrect) and current sufficient memory representations (CorrectCorrect) trials, and to determine subsequent recall success as contrast between later sufficient (ErrorCorrect, CorrectCorrect) and later insufficient (ErrorError) memory representations. CorrectError trials were excluded since they were not present in all participants and because the quality of memory representations for a successful recall was doubtful due to its later memory failure. During memory recall, encoding demand was associated with a 3.2% increase in face-processing evidence $[z(27) = 6.89, p < 0.001, R^2 = 0.637, CI_{95} = 6.52$ to $7.26]$, and subsequent recall success with a 1.1% increase $[z(27) = 1.92, p = 0.055, R^2 = 0.120, CI_{95} = 1.55$ to $2.29]$ in multivariate classification evidence for face representations (Supplementary Table 12). During encoding, encoding demand was estimated to increase the face-processing by 3.9% $[z(27) = 7.18, p < 0.001, R^2 = 0.656, CI_{95} = 6.82$ to $7.55]$ and subsequent recall success was related to a 1.5% $[z(27) = 2.25, p = 0.024, R^2 = 0.158, CI_{95} = 1.88$ to $2.62]$ increased probability in the linear mixed model analysis (Supplementary Table 13). During stimulus inter-trial-interval, neither encoding demand $[z(27) = 1.57, p = 0.116, R^2 = 0.084, CI_{95} = 1.20$ to $1.94]$, nor subsequent recall success $[z(27) = -0.34, p = 0.732, R^2 = 0.004, CI_{95} = -0.71$ to $0.03]$ significantly predicted the single-trial level of evidence for face-processing (Supplementary Table 14). The percentage values in this case mean that, for example, within encoding epochs, there is a 3.9% stronger level of face-processing evidence in trials, which require

improved encoding, such as ErrorError and ErrorCorrect trials, compared to CorrectCorrect trials. On the other hand, face-processing evidence is 1.5% higher on those trials of a participant which are the learning opportunities for subsequent successful recall, i.e., ErrorCorrect and CorrectCorrect trials compared to ErrorError trials. Control analyses restricted to ErrorError and ErrorCorrect trials replicated the association between stimulus processing and subsequent recall success, and ensured that the effect was not driven by the inclusion of CorrectCorrect trials (see Supplementary Tables 15-17). The increase in stimulus representation strength during recall and encoding suggests increased allocation of attention to the presented stimulus category according to the necessity of learning and the success in forming association memories. During a failed recall, the need for increased attention may already become evident and increase face-processing for the following encoding attempt. During encoding, enhanced face-processing indicated a facilitation in memory formation by successful subsequent recall. Increased processing of the memory-relevant stimulus category in the ventral visual stream, therefore, links the monitoring memory errors with improved associative learning both on a behavioral level and related to a network of cognitive control regions.

## Discussion
The present study aimed to investigate which brain regions detect memory errors and coordinate adaptation processes to improve memory formation. Different sources of memory-error evidence overlapped in a pMFC cluster, which showed increased hemodynamic responses during memory-error-related events, such as during failed recall, the selection of low confidence

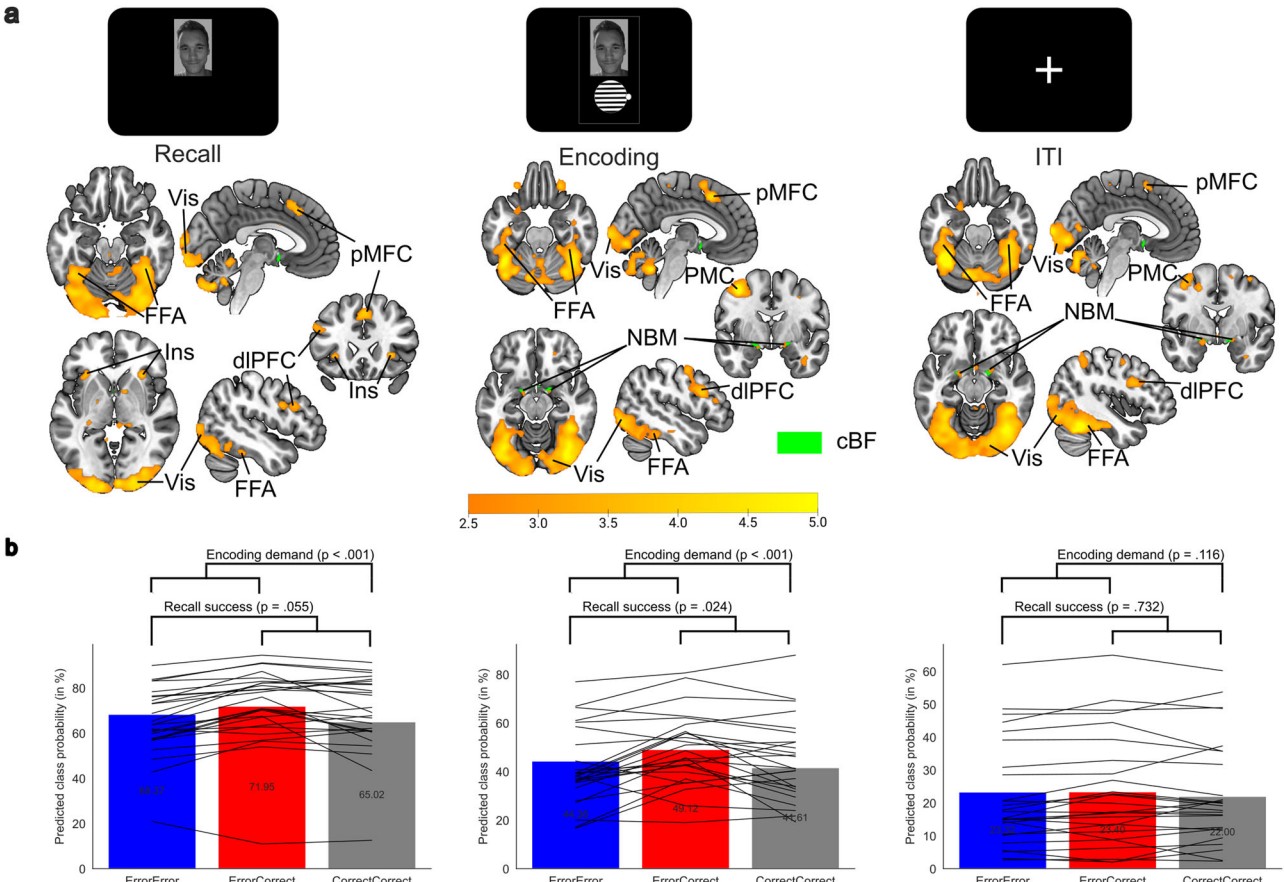

**Fig. 5 | Neurophysiological and behavior underpinnings of FFA-based face-processing evidence during memory-relevant epochs in the feedback-based association learning task.** The support vector machines, which were trained on the 1-back localizer task of each participant ($n = 28$, two participants were excluded from cross-classification analyses based on low localizer task performance), predicted face processing during memory recall, encoding and inter-trial-interval in the feedback-based association learning task, based on the 14 most face-selective voxels in the cytoarchitectonic probability mask of a fusiform gyrus 4 region. **a** General linear model results on evidence for face-processing displayed increased hemodynamic response in regions among fusiform face area (FFA), posterior medial frontal cortex (pMFC), dorsolateral prefrontal cortex (dlPFC), anterior insula (Ins), premotor cortex (PMC), and a cluster overlapping with a cytoarchitectonic mask of the basal nucleus of Meynert (NBM) subregion of the cholinergic basal forebrain (cBF) and amygdala. **b** The level of evidence for face-processing was higher when there was a demand of memory improvement during recall and encoding, and significantly higher for subsequent recall success during encoding epochs, as found in the linear mixed model results.

and the presentation of negative feedback. A posterior portion of the error-related pMFC cluster further distinguished later successfully remembered memory-error trials from repeatedly failed memory formation attempts. The level of FFA-based face-processing evidence was related to increased single-trial hemodynamic responses in a network of cognitive control regions. This network encompassed pMFC, dlPFC, visual cortex, anterior insula and a cluster overlapping with basal forebrain and amygdala, for the upregulation of memory-relevant stimulus representations. Stronger face-processing evidence in the ventral visual stream was linked to the demand of improving memory formation during failed recall, and was further upregulated during improved post-error encoding epochs, as determined by subsequent recall success.

The results favor the perspective that pMFC is involved in monitoring incorrect and low-confident memory representations and that it orchestrates brain networks involved in allocating attention to the relevant stimulus category for error-driven improvements in memory formation. Previous studies have shown that the monitoring of behaviorally relevant events is associated with hemodynamic responses and electrophysiological signals in pMFC (for a review, see[6]). It has been an open question whether pMFC involvement in performance monitoring also applies to evaluating the quality of memory representations. The current study has shown that increased hemodynamic responses in pMFC are related to the processing of negative feedback, as had been described for failed associative recall in

previous fMRI[9] and EEG[10] studies. Furthermore, overlapping clusters in pMFC were also found for hemodynamic responses during failed recall attempts and upon reporting low confidence, which suggests a more general role of pMFC in accumulating evidence of memory errors beyond the processing external error evidence, such as during the presentation of negative feedback. If increased fMRI signals in pMFC are relevant for recognizing the insufficiency of current memory representations, the involvement of pMFC in the post-error subsequent memory effect suggests a role in driving error-following adjustments in associative learning. Results on the post-error subsequent memory effect complement a meta-analysis of previous fMRI studies on the subsequent memory effect[4], which has shown consistent involvement of pMFC in the subsequent memory effect but has not described its function for memory formation. Consistent with previous studies in the cognitive control literature, error-related signals in pMFC have shown to be predictive for later recall success[9,10] and enhanced performance in other cognitive tasks[7,8]. In this regard, the results demonstrate that pMFC is not only related to successful encoding but its hemodynamic responses are already increased upon monitoring error evidence such as during the presentation of negative feedback, which is closely linked to encoding demand and emphasizes a preparatory function for following learning attempts.

While previous studies have shown increased pMFC-based error signals for improved performance, it has been an open question how post-error

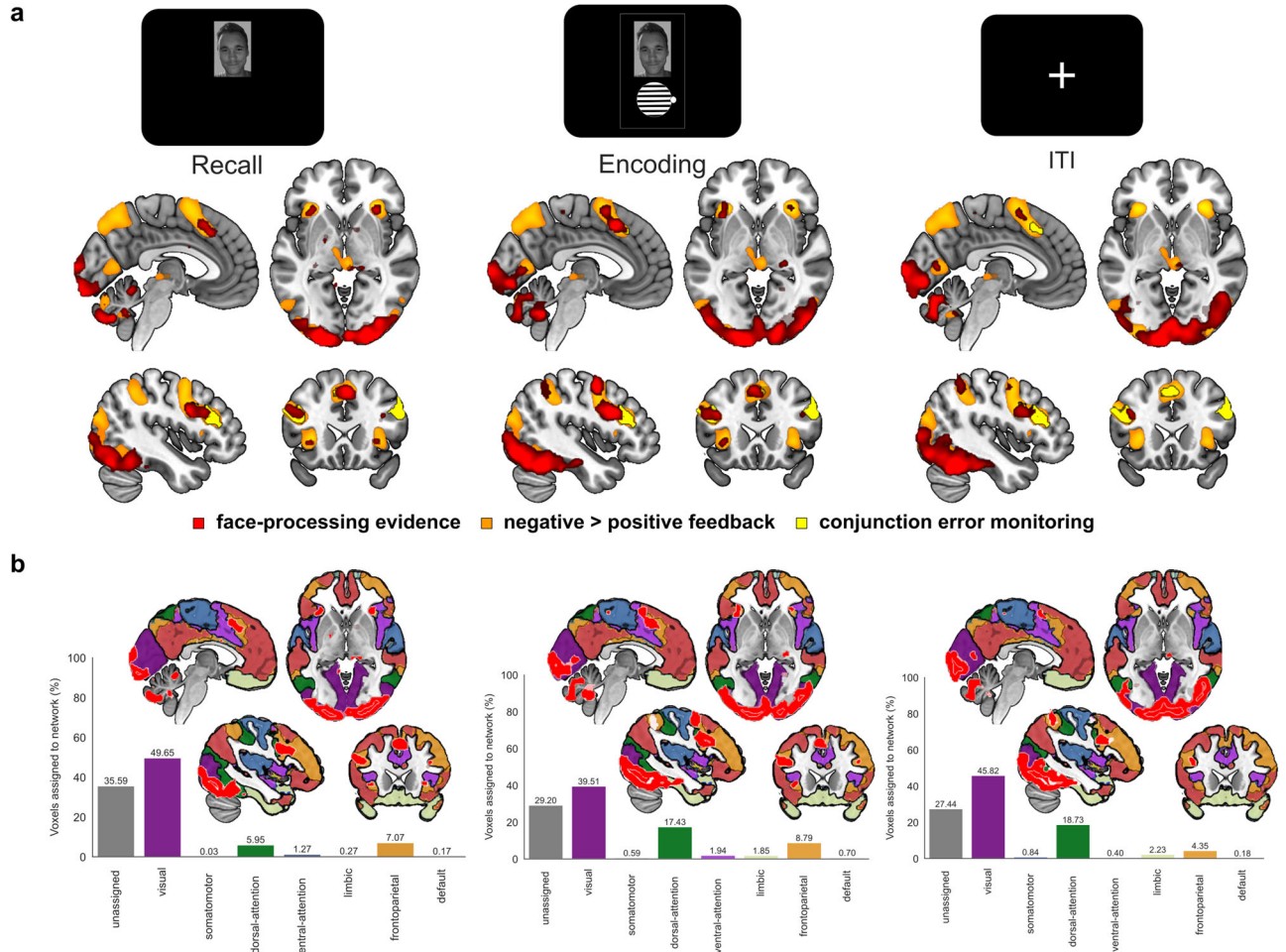

**Fig. 6 | Neurophysiological associations of face-processing evidence in face-specific regions and the similarity with the hemodynamic topography of error monitoring processes and intrinsic brain networks. a** Overlap between the group level (n = 28, two participants were excluded from cross-classification analyses based on low localizer task performance) hemodynamic topography associated with increased face-processing evidence (red), the conjunction of memory-error monitoring processes (yellow) and negative feedback (orange). **b** Network correspondences of significant voxels in respective contrasts related to face-processing evidence during recall, encoding and inter-trial-interval epochs in the feedback-based association learning task. Most voxels were assigned to the visual network, followed by frontoparietal and dorsal attention networks with a lower proportion.

learning improvements are implemented. One of the speculated mechanisms how failed recall leads to enhanced memory formation, has been increased attentional allocation[18]. The current study tested the hypothesis that detected recall errors increase the processing of memory-relevant stimulus representations to facilitate association learning. FFA-based fMRI evidence for face-category processing was used as a proxy measure for stimulus-based attention and showed that hemodynamic responses in regions such as pMFC, dlPFC, anterior insula and the basal forebrain increase as a function of stimulus-specific processing evidence. These regions may interact to enhance attention for following learning attempts. While multivariate pattern analyses have been used to estimate levels of attention, it remained to be shown that a marker for allocated attention provides a link between memory-error detection and improved learning. Previous studies have used multivariate fMRI analyses to show that decoding accuracies and classification probabilities are increased for attended objects. More specifically, the highest decoding accuracies of occipitotemporal stimulus representations have been found for stimuli in the focus of attention[21] and when they are behaviorally relevant[19]. Another study used a combination of multivariate classification probabilities and eye tracking to develop a marker for how much attention was allocated[20]. By using single-trial stimulus class probabilities instead of binary classification accuracies[22], the relationship between neurophysiological processing strength of memory-relevant stimulus representations and their behavioral

correspondence to encoding demand and subsequent recall success became apparent. This suggests that multivariate evidence for stimulus-processing during associative learning can be used as a marker for stimulus-based attention and represents a link between performance monitoring and improved memory formation. The current study aligns with previous studies linking multivariate stimulus models with behavior, by showing that single-trial evidence for face-processing in face-selective ventral visual stream regions is associated with increased hemodynamic responses in pMFC. This suggests a systematic relationship between the neurophysiological underpinnings of enhanced stimulus representations, the detection of memory errors and following encoding success.

Assuming that, upon the detection of respective task demands, pMFC upregulates stimulus-selective regions such as FFA for face processing, direct or indirect synaptic connections between these regions could mediate error-driven adaptations on visual attention[37]. Rodent studies suggested that direct connections between midfrontal and visual regions underly post-error upregulation of visual attention[38]. Other studies emphasized that lateral frontoparietal network regions, such as dlPFC, are responsible for maintaining stimulus representations for memory formation[21,39]. In the current study, representation strength was also associated with increased hemodynamic responses in dlPFC, suggesting it as an important node of a control network for attentional allocation. The current study quantified the proportion of voxels corresponding to particular intrinsic networks[14] within

respective results of error monitoring processes, memory formation and face-processing evidence. While in the error monitoring conjunction the majority of voxels was assigned to the lateral frontoparietal control network, in the post-error subsequent memory effect the ventral and dorsal attention networks were primarily involved. In relation to face-processing evidence, the majority of voxels was assigned to the visual network, followed by dorsal attention and frontoparietal networks. In this regard, demand-dependent adaptations in stimulus-specific regions, such as FFA in the ventral visual stream, could be modulated by a combination of frontoparietal control, ventral attention and dorsal attention networks. Accordingly, previous studies have hypothesized that the ventral attention network involves the frontoparietal network for external attention allocation[15], which may then have downstream effects on sensory regions such as the visual network. Results from this study partially support this assumption by showing pMFC and dlPFC clusters assigned to the frontoparietal network are consistently upregulated during memory error monitoring processes and in concert with increased stimulus evidence during recall, encoding and inter-trial-interval. In the post-error subsequent memory effect, the pMFC cluster was superior and posterior to the error monitoring conjunction, with a larger proportion of the ventral attention network and an overlap with the contrast for negative feedback. From a large-scale brain network perspective, a better understanding on network interactions, which may start with a consensus on network naming and functions[17], may help disentangle in which mechanistic order ventral attention, frontoparietal and dorsal attention networks are engaged. Interestingly, during encoding and inter-trial-interval, FFA-based evidence for face-processing was also related to a cluster at the border zone to the basal forebrain, a region important for modulating arousal[40,41] and releasing the neuromodulator acetylcholine. The cholinergic system has shown to mediate post-error upregulation of visual attention in a pharmacological fMRI study[42]. Further work is needed to determine to which degree these different pathways are exclusive or working in concert, to understand whether and when an error-driven increase of stimulus processing is caused by direct pMFC connections to stimulus-specific regions, mediated by frontoparietal control network regions such as dlPFC and/or modulated by the basal forebrain cholinergic system.

## Limitations

The current study investigated neurophysiological associations of memory-related demand detection processes, associative learning improvements and memory-relevant stimulus processing evidence. Because of the abundant literature on ventral visual stream regions showing face-specific processing, such as in FFA, the study was designed to assess face-processing evidence during associative memory formation based on relevant cytoarchitectonic masks of the fusiform gyrus. In this regard, face stimuli were the presented memory cues which were to be associated with different target orientations of gabor patches. While the study showed an error-driven and subsequent recall success-related upregulation of face processing in the fusiform gyrus, respective analyses on the target orientations of gabor patches were not performed. Although previous studies were able to decode orientations especially from visual cortical regions, the low number of repetitions per particular orientation of respective gabor patches ($n = 7$) prevented us from fitting robust machine learning models. Therefore, it cannot be concluded that the same regions upregulate the processing of association-memory cue and target stimuli. Further studies are needed to assess the generalizability and robustness of memory-relevant cognitive control regions and their replicability for different stimulus categories.

The results of this study showed that the post-error subsequent memory effect contrast was overlapping with a posterior, superior portion of pMFC as found in the contrast related to negative feedback processing, but not to the exact location of the error-monitoring conjunction per se. Overall, pMFC overlaps with the processing of negative feedback were abundant also in relation to increased face processing evidence. While negative feedback represents the most explicit evidence for required learning improvements and adaptations during the following encoding epoch, it was not the primary purpose of the study to differentiate sources of memory-error evidence

accumulation processes, such as internal evidence from confidence levels and external evidence from negative feedback. Control analyses suggested sufficiently low levels of multicollinearity to assess respective contrasts and different epochs, potentially based on self-paced motor responses in the selection periods during task performance which separate different cognitive events within the same trials. However, adapted task designs will be better suited to investigate how different origins of memory-error evidence differ in their hemodynamic topography. Future studies addressing this research question may benefit from assessing inter-individual variations in paracingulate gyrification patterns to reveal under which circumstances which pMFC subregions are associated with particular performance monitoring processes and large-scale brain networks, jointly or specifically.

Previous studies conducting multivariate fMRI analyses have often used classification accuracies based on a binary classification on a given trial instead of more fine-grained parametric measures, such as class probabilities[22] or the distance from a multivariate hyperplane. Here, we applied multivariate cross-classification analyses to investigate whether and how the level of face-processing evidence in an associative learning task is varying in relation to detected task demands and successful memory formation adaptations. While the localizer task and respective training of the multivariate models contained the same number face and house stimuli, in the feedback-based association learning task only face stimuli were used as memory cues. Accordingly, classification accuracies in the feedback-based association learning task should be interpreted as the relative face-processing evidence and classification rates below 50% do not reflect chance-level performance. Validity of the model was confirmed by robust cross-validation with balanced accuracies in the localizer task, a systematic inter-individual link between cross-classification prediction rates and the average scaled face-class probability, as well as replicable hemodynamic topographies associated with face processing evidence during different memory-relevant epochs. Most importantly, external validity of the model's scaled probability function was indicated by the separation of behavioral relevance measures, such as encoding necessity and subsequent recall success. Overall, quantifying the relative level of face-processing evidence instead of applying binary classification accuracies enabled us to link trial-by-trial variations in encoding demand to the recruitment of cognitive control processes.

## Conclusions

The current study showed that higher hemodynamic responses in pMFC are not only related to improved encoding but are already increased when there is evidence of currently insufficient memory representations. Higher FFA-based face-processing evidence was accompanied by a systematic increase of hemodynamic responses in regions among pMFC, dlPFC, visual cortex, anterior insula, basal forebrain and amygdala. When sufficient evidence on memory errors has been detected, these regions may interact to increase attention during encoding and improve following learning attempts. In the past years, multivariate fMRI analyses have gained popularity and decoding accuracies of brain regions have been used as an estimate for how much stimulus information is represented in neurophysiological data. In this regard, the current study highlights how multivariate stimulus-based models vary in correspondence with hemodynamic responses of the pMFC region with frontoparietal and ventral attention network contributions, which may monitor task demands and detect memory errors. The results help explain in correspondence with which brain regions stimulus representations are enhanced for improved memory formation, and emphasize memory-error detection as a basis for adaptive task performance and associative learning. Future studies may implement single-trial analyses and investigate multivariate processing evidence, to explain why memory formation fails or succeeds from time to time.

## Data availability

The behavioral data, as well as unthresholded and thresholded statistical fMRI maps used for generating the figures are available online at https://osf.io/yr6n2/.

## Code availability

The code used for behavioral and fMRI analyses is available online at https://osf.io/yr6n2/.

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

## Acknowledgements

This work was supported by the German research foundation (DFG), Project 13 in the Research Training Group GRK 2413 "The ageing synapse" (SynAGE). The funders had no role in study design, data collection and analysis, decision to publish or preparation of the manuscript. We thank

Christina Becker, Kathleen Rödger and Halla Mulla-Osman for helping in the recruitment of the participants.

## Author contributions

CRediT author statement: Alexander Weuthen: Conceptualization, Software, Investigation, Data Curation, Writing - Original Draft, Writing - Review & Editing, Project administration. Hans Kirschner: Conceptualization, Software, Writing - Review & Editing. Markus Ullsperger: Conceptualization, Writing - Review & Editing, Project administration, Funding acquisition.

## Funding

## Competing interests

The authors declare no competing interests.
