## [Transparent Peer Review file · Communications Psychology]

Error-driven upregulation of memory representations

Corresponding Author: Dr Alexander Weuthen

Version 0:

Decision Letter:

Dear Dr Weuthen,

Thank you for your patience during the peer-review process. Your manuscript titled "Error-driven upregulation of memory representations" has now been seen by 2 reviewers, and I include their comments at the end of this message. They find your work of interest but raised some important points. We are interested in the possibility of publishing your study in Communications Psychology, but would like to consider your responses to these concerns and assess a revised manuscript before we make a final decision on publication.

We therefore invite you to revise and resubmit your manuscript, along with a point-by-point response to the reviewers. Please highlight all changes in the manuscript text file.

Editorially, the following issues are considered key for your revision.

First, the reviewers raise multiple methodological concerns (especially Ref#1: 1c & 3; Ref #2: 5-7) We ask that your revision includes additional analyses that will shed light on whether the referees' concerns can be alleviated, rather than responding only via textual revisions.

Second, we ask that you ensure that your revision follows the journal's standards for the reporting and interpretation of statistics (see attached Editorial Request Table). In particular, claims about the absence of an effect/an association or claims that an effect/association is specific to another condition require positive evidence for the null in the form of Bayesian statistics or equivalence tests. Differences between contrasts (specificity claims) cannot be presented rhetorically, without at least confirming evidence from interaction contrasts. Finally, we ask that while the referees' requests for an improved discussion of the results should be followed, please keep reverse inference and speculation on mechanism to a minimum.

I am attaching an Editorial Requests Table that details critical reporting requirements for the revised manuscript. Please attend to each item and ensure your manuscript is fully compliant. We are requesting that your manuscript aligns with these requirements as this facilitates the evaluation of your manuscript, reducing delays in re-review and potential future acceptance. If your revised manuscript is not aligned with these requests on major issues, such as those concerning statistics, it may be returned to you for further revisions without re-review. Additional information can be found in our style and formatting guide <https://www.nature.com/documents/commspsychol-style-formatting-guide-accept.pdf>>Communications Psychology formatting guide.

Please use the following link to submit your

- revised manuscript,
- point-by-point response to the referees' comments,
- cover letter (as a separate document),
- the Editorial Policy Checklist (see below),
- the Reporting Summary (see below), and
- the completed Editorial Request Table (attached):

Link Redacted

Best regards,

Marike

Marike Schiffer, PhD
Chief Editor
Communications Psychology

REVIEWER EXPERTISE:

Reviewer #1 cognitive control, fMRI
Reviewer #2 cognitive control, fMRI

REVIEWER REPORTS:

Reviewer #1 (Remarks to the Author):

The authors investigated neural underpinnings of the subsequent memory effect (SME) for incidental learning of faces following classification errors on an unrelated task. Using fMRI, they reveal a common pattern of activity, primarily focusing on performance monitoring (posterior medial frontal cortex) and stimulus-specific (facial fusiform area) regions. While pMFC was broadly involved in monitoring activities, classification of FFA activation revealed enhanced network activity (including FFA and pMFC) during memory recall and encoding. In total, the data were compelling and the findings were reasonably derived from this evidence. I have a few comments that I think may improve the communication of the paper for a broader audience.

Comments

- 1) The writing, particularly in the introduction, is a little stilted and the message is conveyed unevenly. More specific issues about the final stage of analyses are prevalent as well:
 - a) It was unnecessarily difficult to understand the total nature of the experimental design, primarily the classification task and application to FALT performance.
 - b) In the results, the description of the 1-back task for stimulus classification takes a little too long to reveal its purpose.
 - c) All of page 16 could be clearer on what role the correct-correct trials are playing in this analysis: it seems like the authors are not clearly dividing analyses from standard vs. error-driven network activations.
 - d) Lines 275-276 ("predicted class probability averages") is similarly opaque - is this just individual differences in accuracy scaling with out-of-sample transfer identification?
- 2) What does it mean that recall and encoding were associated with increased FFA activity, while rehearsal was not? What is special about rehearsal here?
- 3) There is a lot of discussion of 'overlap' in pMFC (line 163, 204, 291), which makes the absence of a conjunction image surprising. A conjunction or overlap image across Fig 2b,c,d would be beneficial, as would one for Fig 2 and Fig 3b, and one showing the overlap between Figure 2 conditions and Figure 5 conditions.

Reviewer #2 (Remarks to the Author):

The primary objective of this paper is to identify brain regions that are crucial for monitoring encoding demands and to explore how this monitoring process enhances attention to materials that need to be remembered. The study introduces a novel feedback-based associative learning task, with a specific focus on the role of the posterior medial frontal cortex (pMFC). It also employs an FFA-based multivariate classification technique to establish a link between improved memory performance and increased attention. The design of the task is thoughtful and the analytical methods used are advanced,

contributing significantly to our understanding of cognitive processes. However, the study could benefit from greater precision in the use of the term 'pMFC,' which may encompass a broad area belonging to various intrinsic networks. Similarly, terms like 'a cognitive control network' lack anatomical clarity and should be more explicitly tied to prior research. Some results are ambiguously presented, requiring further clarification to enhance understanding. Below are my detailed comments:

1. **pMFC and Intrinsic Networks:** The delineation between the pMFC regions associated with error-monitoring (Figure 2b, c, d) and encoding success (Figure 3b) is crucial, as these may represent subregions of different intrinsic networks. The first might belong to the salience network or ventral attention network, and the latter to the frontoparietal control network. However, the paper ambiguously suggests these regions overlap in function. For clarity, the discussion should explicitly address whether these are distinct regions supporting different functions within the broader context of known intrinsic networks, using precise location data from the results.
2. **FFA-based Face-processing and Associated Activity:** The regions shown in Figure 5b, associated with FFA-based face-processing, are described within a cognitive control network framework. Considering their topography appears to integrate elements of the frontoparietal control, dorsal attention, and visual networks, a discussion that acknowledges this synthesis could offer novel insights and refine our understanding of network interactions during cognitive tasks.
3. **Hippocampal Involvement in Memory:** The significant involvement of the hippocampus in memory formation is underscored by Supplementary Tables 1, 5, 8, and 9. However, the paper lacks a discussion on how attention may influence hippocampal activation. Addressing this could resolve outstanding questions and enhance the paper's contribution to the literature on memory systems.
4. **Task Design and Memory Encoding:** The study focuses on an associative memory task linking faces with Gabor patches, yet the discussion primarily addresses the encoding of faces, neglecting the associative aspect of the task. This oversight could mislead interpretations of the task's requirements and its impact on memory performance. A more balanced discussion of both components—face and association—would align better with the task's design and objectives.
5. **Statistical Dependence Across Epochs:** The paper examines five types of epochs—Recall, Confidence, Feedback, Encoding, and ITI—which correspond to distinct phases of the trials. The potential for statistical dependence among these epochs warrants discussion. It would be beneficial for the authors to explain how they managed this in their experimental design and to acknowledge any limitations this might impose on the study's findings.
6. **Classification Rates Explanation:** On page 13, the reported classification rates for face presentation during different epochs are notably low (39.53% during encoding and 17.35% during rehearsal). These figures require clarification regarding their implications. Are these rates considered successful, or do they indicate limitations in the classifier's performance? An explanation would aid in understanding the effectiveness of the employed methods.
7. **Clarification of Face-processing Evidence Statistics:** On page 16, the reported increases in face-processing evidence (3.2%) and recall success (1.1%) derived from multivariate classification evidence need clarification. Are these increases indicative of improvements in classification accuracy? Detailing this would help in interpreting the statistical significance and practical implications of these findings.

EDITORIAL POLICIES

We ask that you ensure your manuscript complies with our editorial policies and reporting requirements.

To that end, we require revised manuscripts to be accompanied by two completed items: a reporting summary that collects information on study design and procedure, and an editorial policy checklist that verifies compliance with all required editorial policies.

- <https://www.nature.com/documents/nr-reporting-summary.zip>>Nature Research Reporting Summary
- <https://www.nature.com/documents/nr-editorial-policy-checklist.pdf>>Editorial Policy Checklist

All points on the policy checklist must be addressed. Your revised manuscript can only be sent back to the referees if these checklists are completed and uploaded with the revision.

Notes: If you have submitted a Stage 1 Registered Report, Review, Primer, Comment, or Perspective you do not need to submit these forms. If you have already submitted these forms, you may disregard this request.

Version 1:

Decision Letter:

Dear Dr Weuthen,

Your manuscript titled "Error-driven upregulation of memory representations" has now been seen by our reviewers, whose comments appear below. In light of their advice I am delighted to say that we are happy, in principle, to publish a suitably revised version in Communications Psychology.

We therefore invite you to revise your paper one last time to address the remaining concerns of our reviewers and a list of editorial requests. At the same time we ask that you edit your manuscript to comply with our format requirements and to maximise the accessibility and therefore the impact of your work.

EDITORIAL REQUESTS:

SUBMISSION INFORMATION:

OPEN ACCESS:

* **DATA AVAILABILITY:**

Link Redacted

Best regards,

Marika

Marika Schiffer, PhD
Chief Editor
Communications Psychology

REVIEWERS' COMMENTS:

Reviewer #1 (Remarks to the Author):

The authors have addressed all of my rather minor concerns with this revision.

Reviewer #2 (Remarks to the Author):

I commend the authors for their thorough responses to my queries, including clarifying the association with specific intrinsic networks, outlining the study design's limitations, and providing a clearer explanation of the classification outcomes. I have no further questions at this time.

We thank the Editor for arranging these helpful reviews and we thank the reviewers for their helpful comments on the manuscript. We have addressed the raised concerns in point-by-point responses. We conducted further analyses and provided respective figures and results, which have been added to the manuscript and, in part, to the supplementary materials. Several sections have been rewritten to convey the message more clearly, and the manuscript format has been adjusted to comply with the journal's guidelines. Additionally, a limitations section has been added to address key discussion points raised by the reviewers.

Reviewer #1

Remarks to the Author:

The authors investigated neural underpinnings of the subsequent memory effect (SME) for incidental learning of faces following classification errors on an unrelated task. Using fMRI, they reveal a common pattern of activity, primarily focusing on performance monitoring (posterior medial frontal cortex) and stimulus-specific (facial fusiform area) regions. While pMFC was broadly involved in monitoring activities, classification of FFA activation revealed enhanced network activity (including FFA and pMFC) during memory recall and encoding. In total, the data were compelling and the findings were reasonably derived from this evidence. I have a few comments that I think may improve the communication of the paper for a broader audience.

We thank the reviewer for the interest in our work. The raised points helped us to clarify the analytical strategy and will aid the reader in understanding the conclusions from the manuscript.

Comments

1) The writing, particularly in the introduction, is a little stilted and the message is conveyed unevenly. More specific issues about the final stage of analyses are prevalent as well:

a) It was unnecessarily difficult to understand the total nature of the experimental design, primarily the classification task and application to FALT performance.

We agree that the design of the task and cross-classification analyses were not easy to understand. We have revised and simplified the explanations of our approach in both the Introduction and Methods sections to make the manuscript clearer and more concise.

b) In the results, the description of the 1-back task for stimulus classification takes a little too long to reveal its purpose.

The results section on the localizer task has been adapted and is now written more concisely. In particular, we removed details which can be sufficiently derived from the methods section.

Results (p. 18 l. 436): The 1-back localizer task captured face-selective processing in a cytoarchitectonic mask of the fusiform gyrus

To build a model of stimulus representation strength, we trained a classifier to distinguish fMRI data during face and house stimulus presentations in a 1-back localizer task (Fig. 4a). 28 out of 30 participants performed the task either without mistakes or within two standard deviations from the group average (Fig. 4b).

Univariate GLM analyses in the 1-back localizer task showed that hemodynamic responses were larger for faces than houses in FFA, as determined by a strong overlap with cytoarchitectonic probability maps of left and right FG-4

[$\text{Face}_{\text{CorrectRejection}} > \text{House}_{\text{CorrectRejection}}$; $z(27) = 5.48$, $p_{\text{FDR}} < .001$, $R^2 = .527$, $\text{CI}_{95} = 5.11$ to 5.85 ; $x = 44$, $y = -46$, $z = -27$], but also in other regions previously described as face-selective such as superior temporal sulcus [$z(27) = 4.48$, $p_{\text{FDR}} < .001$, $R^2 = .426$, $\text{CI}_{95} = 4.11$ to 4.85 ; $x = 51$, $y = -46$, $z = 5$] and anterior temporal lobe [$z(27) = 4.68$, $p_{\text{FDR}} < .001$, $R^2 = .448$, $\text{CI}_{95} = 4.31$ to 5.05 ; $x = 40$, $y = 19$, $z = -31$, Fig. 4 and Supplementary Table 6]. Increased hemodynamic responses for houses compared to faces were found in regions among parahippocampal gyrus [$\text{House}_{\text{CorrectRejection}} > \text{Face}_{\text{CorrectRejection}}$; $z(27) = 7.02$, $p_{\text{FDR}} < .001$, $R^2 = .646$, $\text{CI}_{95} = 6.65$ to 7.39 ; $x = -29$, $y = -52$, $z = -5$; Fig. 4 and Supplementary Table 7]. Overall, univariate fMRI results in the localizer task displayed the classical dissociation in the ventral visual stream, displaying FFA-related hemodynamic responses being larger for faces and house-specific hemodynamic responses in parahippocampal gyrus.

In the localizer task, a machine learning model was trained, in order to predict the strength of FFA-based face-processing evidence during memory-relevant epochs in FALT. Leave-one-run-out cross-validation reached an average balanced decoding accuracy of 73.15 % [$t(27) = 12.38$, $p < .001$, one-sample t-test, $> 50\%$ chance level, $R^2 = .855$, $\text{CI}_{95} = 69.96$ to 76.33 %] on distinguishing faces and houses based on 14 ANOVA-feature selected voxels within FFA. The prediction of house and face stimuli was balanced, showing no trend in the likelihood of the face representation strength models to prefer either of both categories [$t(27) = -0.18$, $p = .855$, two-sample t-test, $R^2 = .001$, $\text{CI}_{95} = -.002$ to $-.003$]. Taken together, the decoding accuracies and control analyses suggested that the multivariate face-processing model was able to evaluate face-processing evidence by distinguishing face and house trials.

c) All of page 16 could be clearer on what role the correct-correct trials are playing in this analysis: it seems like the authors are not clearly dividing analyses from standard vs. error-driven network activations.

We thank the reviewer for pointing this out. We have adapted the section and added a justification on the inclusion of CorrectCorrect trials for understanding the origin of changes in the level of stimulus evidence according to both the necessity on and success in encoding. Please also see our response to Reviewer #2 comment 7 for more clarification on the statistical effects and possible interpretations based on the inclusion of CorrectCorrect trials.

Results (p. 21 l. 513): The proxy measure for the strength of stimulus representations, as a marker of allocated attention to the presented stimulus category, was also analyzed regarding its correspondence with behavioral necessity and success on learning the presented associations (Fig. 5b). Previous fMRI GLM analyses indicated increased hemodynamic responses in pMFC both for error-monitoring processes related to encoding demand and subsequent memory

performance related to encoding success. A link between the level of stimulus representations strength and both, encoding necessity and success, remained to be assessed. The analyses were restricted to ErrorError, ErrorCorrect and CorrectCorrect trials to determine encoding demand as contrast between current insufficient (ErrorError, ErrorCorrect) and current sufficient memory representations (CorrectCorrect) trials, and to determine subsequent recall success as contrast between later sufficient (ErrorCorrect, CorrectCorrect) and later insufficient (ErrorError) memory representations.

To ensure that the relationship between encoding success and face-processing evidence is not driven by the inclusion of the CorrectCorrect trials, we performed additional linear mixed model analyses to directly assess the difference between ErrorError and ErrorCorrect trials for their effects on stimulus representation strength. Results of the adapted analyses for all three epochs can be found in the supplementary tables 13-15. The results suggest that the effect related to subsequent recall success is not driven by the inclusion of CorrectCorrect trials. We have added this point in the manuscript results section to report the robustness of the effects.

Results (p. 22 l. 539): Control analyses restricted to ErrorError and ErrorCorrect trials replicated the association between stimulus processing and subsequent recall success, and ensured that the effect was not driven by the inclusion of CorrectCorrect trials.

d) Lines 275-276 (“predicted class probability averages”) is similarly opaque - is this just individual differences in accuracy scaling with out-of-sample transfer identification?

We thank the reviewer for highlighting the need for further clarification on this point. For a better understanding, we added a supplementary figure to show the relationship between both variables and elaborated on the meaning and intention to include this step of validating the measure. We hope that our approach is now clearer. Please also see our response to Reviewer #2 points 6 and 7, where we discuss the interpretability, validity and limitations of the measure and corresponding manuscript adaptations in more detail.

Supplementary Figure 2 | Relationship between the average rate of face predictions per participants and average single-trial face-class probability. To validate the cross-

classification-based single-trial stimulus evidence parameter, inter-individual associations were assessed. Cross-classification prediction rates and average class probability showed a strong association (Spearman correlation $> .90$) during all three epochs, a, recall, b, encoding and c, ITI.

Limitations (p. 29 l. 712): Previous studies conducting multivariate fMRI analyses have often used classification accuracies based on a binary classification on a given trial instead of more fine-grained parametric measure such as class probabilities²² or measures such as the distance from a multivariate hyperplane. Here, we applied multivariate cross-classification analyses to investigate whether and how the level of face-processing evidence in an associative learning task is varying in relation to detected task demands and successful memory formation adaptations. While the localizer task and respective training of the multivariate models contained the same number face and house stimuli, in FALT only face stimuli were used as memory cues. Accordingly, classification accuracies in the FALT should be interpreted as the relative face-processing evidence and classification rates below 50% do not reflect chance level performance. Validity of the model was confirmed by robust cross-validation with balanced accuracies in the localizer task, a systematic inter-individual link between cross-classification prediction rates and the average scaled face-class probability, as well as replicable hemodynamic topographies associated with face processing evidence during different memory-relevant epochs. Most importantly, external validity of the model's scaled probability function was indicated by the separation of behavioral relevance measures such as encoding necessity and subsequent recall success. Overall, quantifying the relative level of face-processing evidence instead of applying binary classification accuracies enabled us to link trial-by-trial variations in encoding demand to the recruitment of cognitive control processes.

2) What does it mean that recall and encoding were associated with increased FFA activity, while rehearsal was not? What is special about rehearsal here?

We thank the reviewer for pointing out the need to address the specificity of stimulus rehearsal. We initially expected a link between a behaviorally-relevant increase in rehearsal and FFA-based face representation strength. However, from the design of the task we have now reconsidered that it cannot be inferred neither which participants, nor in which trials particular participants have used the inter-trial-interval (ITI) for stimulus rehearsal. We have, therefore, reduced the emphasis on rehearsal processes within the ITI, both in the figures and manuscript. Due to the heterogeneity of potential accompanying processes, we want to refrain from further speculations on rehearsal-specific dynamics.

3) There is a lot of discussion of 'overlap' in pMFC (line 163, 204, 291), which makes the absence of a conjunction image surprising. A conjunction or overlap image across Fig 2b,c,d would be beneficial, as would one for Fig 2 and Fig 3b, and one showing the overlap between Figure 2 conditions and Figure 5 conditions.

We thank the reviewer for pointing out the need to assess topographical overlaps more specifically. We performed a conjunction analysis on the error monitoring contrasts and included the conjunction image in Fig. 2. Overlaps with the results

from the post-error subsequent memory effect and cross-classification analyses are also included now in Fig. 3 and Fig. 5, respectively. We have further adapted the methods, results and discussion sections to include the description and implications from the conjunction and contrast overlaps.

Fig. 2 | (...) e, The conjunction image shows voxels which had significant hemodynamic responses in all three memory error monitoring contrasts (failed recall, low confidence, negative feedback). The effects converged in pMFC, right premotor cortex (PMC) and bilateral dorsolateral prefrontal cortex (dIPFC).

Fig 3 | (...) c, Overlap of the post-error SME (red) with the conjunction of error monitoring contrasts (yellow) and the contrast negative feedback (orange). The pMFC cluster for the post-error SME overlapped with a posterior portion of the cluster related to processing negative feedback, suggesting that its demand-dependent upregulation may have a preparatory function.

Fig. 6 | Neurophysiological associations of face-processing evidence in face-specific regions and the similarity with the hemodynamic topography of error monitoring processes and intrinsic brain networks. a, Overlap between the hemodynamic topography associated with increased face-processing evidence (red), the conjunction of memory-error monitoring processes (yellow) and negative feedback (orange).

Methods (p. 11 l. 267): To assess in which brain regions memory error monitoring processes converge, a conjunction analysis was performed by identifying voxels with significant effects in all contrasts i.e., failed recall, a low level of confidence and negative feedback. The conjunction effects were statistically tested against a chance level of $p < .05$ based on 10000 random permutations³⁵. To show overlaps, such as between memory error monitoring-related hemodynamic responses and the post-error subsequent memory effect, the statistical results were overlaid with the conjunction image and plotted together with the contrasts for negative feedback as the largest topographical extent and most explicit evidence level on memory errors.

Results (p.17 l. 400): A conjunction analysis on all three error monitoring contrasts showed overlaps in a pMFC region encompassing both hemispheres, bilateral dorsolateral prefrontal cortices (dlPFC) and right premotor cortex (PMC). Most of the significant voxels (> 90 %) in the error monitoring conjunction were assigned to the frontoparietal control network.

(p. 17 l. 417): The location of the post-error SME contrasts was posterior and superior of the conjunction image for memory error monitoring, where it overlapped with the contrast for negative compared to positive feedback.

(p. 20 l. 493) The pMFC topography related to face-processing evidence during recall and encoding, but not during the ITI, overlapped with the error-monitoring conjunction. A posterior and superior pMFC cluster was found during encoding and the ITI, but not during recall. A similar pattern was apparent in dlPFC, showing a more posterior dlPFC cluster during encoding and ITI. However, both in pMFC and dlPFC the effects related to face-processing evidence were consistently overlapping with the hemodynamic response for negative feedback processing in all three epochs. These overlaps showed a shared topography between processes for the

monitoring of inaccurate memory representations and upregulated stimulus representations.

Limitations (p. 28 l. 693): The results of this study showed that the post-error SME contrast was overlapping with a posterior, superior portion of pMFC as found in the contrast related to negative feedback processing, but not to the exact location of the error-monitoring conjunction per se. Overall, pMFC overlaps with the processing of negative feedback were abundant also in relation to increased face processing evidence. While negative feedback represents the most explicit evidence for required learning improvements and adaptations during the following encoding epoch, it was not the primary purpose of the study to differentiate sources of memory-error evidence accumulation processes, such as internal evidence from confidence levels and external evidence from negative feedback. Control analyses suggested sufficiently low levels of multicollinearity to assess respective contrasts and different epochs, potentially based on self-paced motor responses in the selection periods during task performance which separate different cognitive events within the same trials. However, adapted task designs will be better suited to investigate how different origins of memory-error evidence differ in their hemodynamic topography. Future studies addressing this research question may benefit from assessing inter-individual variations in paracingulate gyrification patterns to reveal under which circumstances which pMFC subregions are associated with particular performance monitoring processes and large-scale brain networks, jointly or specifically.

Reviewer #2

Remarks to the Author:

The primary objective of this paper is to identify brain regions that are crucial for monitoring encoding demands and to explore how this monitoring process enhances attention to materials that need to be remembered. The study introduces a novel feedback-based associative learning task, with a specific focus on the role of the posterior medial frontal cortex (pMFC). It also employs an FFA-based multivariate classification technique to establish a link between improved memory performance and increased attention. The design of the task is thoughtful and the analytical methods used are advanced, contributing significantly to our understanding of cognitive processes. However, the study could benefit from greater precision in the use of the term 'pMFC,' which may encompass a broad area belonging to various intrinsic networks. Similarly, terms like 'a cognitive control network' lack anatomical clarity and should be more explicitly tied to prior research. Some results are ambiguously presented, requiring further clarification to enhance understanding. Below are my detailed comments:

We thank the reviewer for the interest in our work and the opportunity to clarify topographical information and network correspondences. We elaborated on the parts of the manuscript, where we discuss the choice for referring to pMFC as a broad area and we performed additional analyses to investigate network correspondences. Please find more details within our responses to the comments below.

1. pMFC and Intrinsic Networks: The delineation between the pMFC regions associated with error-monitoring (Figure 2b, c, d) and encoding success (Figure 3b) is crucial, as these may represent subregions of different intrinsic networks. The first might belong to the salience network or ventral attention network, and the latter to the frontoparietal control network. However, the paper ambiguously suggests these regions overlap in function. For clarity, the discussion should explicitly address whether these are distinct regions supporting different functions within the broader context of known intrinsic networks, using precise location data from the results.

We agree with the reviewer about the necessity to assess and discuss the overlap between error-monitoring contrasts and hemodynamic responses during successful encoding. As requested by Reviewer #1, we performed a conjunction analysis on the error-monitoring contrasts, and assessed overlaps of the conjunction with the subsequent memory and face-processing contrasts. For more details to this point including an added limitation section, please see our response to Reviewer #1 comment 3, where we show respective overlaps and provide the figures within this document. We also added a more detailed section on the anatomical specificity of pMFC in the introduction. We hope that the more elaborated explanation helps explain the choice of using the labeling pMFC despite more fine-grained parcellation schemes.

Introduction (p. 3 l. 57): The pMFC region can be parcellated into more fine-grained subregions such as anterior and posterior midcingulate, (pre-)supplementary motor and dorsomedial prefrontal cortices based on criteria such as cytoarchitectonic profiles ^{11, 12}. These regions have been assigned to contribute to large-scale brain networks, such as a midcingulo-insular salience/ ventral attention network, a lateral frontoparietal/ executive control network and a medial frontoparietal default mode network ^{13, 14}. The ventral attention network has been proposed to switch between the frontoparietal control network for external attention and upregulated default mode network for internal attention ¹⁵. However, precise mapping of functional representations in the pMFC onto its anatomical subregions has proven difficult in human fMRI research. This is partly driven by substantial interindividual variability of pMFC anatomy ¹⁶. Therefore, we refer to the pMFC as a broad region related to performance monitoring processes, which may be assigned to different large-scale brain networks ^{14, 17}.

To address which of the intrinsic networks are most strongly involved, we performed additional analyses regarding network correspondences and quantified the number of overlapping voxels with the major seven intrinsic networks as proposed by Yeo et al. (2011). The procedure has been added to the methods section and we provided additional figures. We also adapted the introduction und discussion sections where intrinsic brain network effects were addressed. Please find the discussion section in our response to the next comment.

Fig 2. | (...) e, The conjunction image shows voxels which had significant hemodynamic responses in all three memory error monitoring contrasts (failed recall, low confidence, negative feedback). The effects converged in pMFC, right premotor cortex (PMC) and bilateral dorsolateral prefrontal cortex (dIPFC). f, Network correspondence was quantified according to the overlap of voxels in the conjunction with a seven-networks cortical parcellation scheme¹⁴. Most voxels of the error monitoring conjunction were assigned to the lateral frontoparietal control network.

d

Fig. 3 | (...) d, The image shows the poster-error SME plotted as overlay to a seven-networks cortical parcellation scheme. Most voxels were assigned to ventral attention network, dorsal attention network and visual network.

Methods (p. 12 | 276): Related to topographical variations in the cluster locations, associations with an often-used seven-networks brain parcellation scheme were assessed, encompassing visual, somatomotor, dorsal attention, ventral attention, limbic, frontoparietal and default networks in the cerebral cortex. Thresholded statistical maps were overlaid to color-coded brain maps for these networks and the relative number voxels assigned to a mask was quantified. Voxels falling outside the cerebral network masks were specified as unassigned.

Results (p. 17 | 403): Most of the significant voxels (> 90 %) in the error monitoring conjunction were assigned to the frontoparietal control network.

(p. 17 | 424): Regarding network correspondences, the post-error SME showed a more variable pattern compared to the error monitoring conjunction contrast. More than 5% of significant voxels were assigned to ventral attention, dorsal attention, visual, default and frontoparietal networks (see Fig. 3d).

2. FFA-based Face-processing and Associated Activity: The regions shown in Figure 5b, associated with FFA-based face-processing, are described within a cognitive control network framework. Considering their topography appears to integrate elements of the frontoparietal control, dorsal attention, and visual networks, a discussion that acknowledges this synthesis could offer novel insights and refine our understanding of network interactions during cognitive tasks.

We thank the reviewer for elaborating on the involved networks found in relation to the level of face-processing evidence. As described in the response to the preceding comment, we now also assessed network correspondences according to the parcellation scheme by Yeo et al. (2011) during the three epochs in Fig. 6. Indeed, as suggested by the reviewer, the strongest topographical overlap is found for visual, dorsal attention and frontoparietal networks. We have adapted the results and discussion sections accordingly, to include how the findings correspond to the literature on large-scale brain networks.

Fig. 6 | (...) b, Network correspondences of significant voxels in respective contrasts related to face-processing evidence during recall, encoding and inter-trial-interval (ITI) epochs in the feedback-based association learning task (FALT). Most voxels were assigned to the visual network, followed by frontoparietal and dorsal attention networks with a lower proportion.

Results (p. 21 l. 502): Assessment of large-scale brain network correspondences indicated the strongest association with the visual network, and to a lesser degree with dorsal attention and frontoparietal networks.

Discussion (p. 26 l. 643): The current study quantified the proportion of voxels corresponding to particular intrinsic networks within respective results of error monitoring processes, memory formation and face-processing evidence¹⁴. While in the error monitoring conjunction the majority of voxels was assigned to the lateral frontoparietal control network, in the post-error SME the ventral and dorsal attention networks were primarily involved. In relation to face-processing evidence, the majority of voxels was assigned to the visual network, followed by dorsal attention and frontoparietal networks. In this regard, demand-dependent adaptations in stimulus-specific regions, such as FFA in the ventral visual stream, could be modulated by a combination of frontoparietal control, ventral attention and dorsal attention networks. Accordingly, previous studies have hypothesized that the ventral attention network involves the frontoparietal network for external attention allocation¹⁵, which may then have downstream effects on sensory regions such as the visual network. Results from this study partially support this assumption by showing pMFC and dIPFC clusters assigned to the frontoparietal network are consistently upregulated during memory error monitoring processes and in concert with increased stimulus evidence during recall, encoding and ITI. In the post-error SME, the pMFC cluster was superior and posterior to the error monitoring conjunction, with a larger proportion of the ventral attention network and an overlap with the contrast for negative feedback. From a large-scale brain network perspective, a better

understanding on network interactions, which may start with a consensus on network naming and functions ¹⁷, may help disentangle in which mechanistic order ventral attention, frontoparietal and dorsal attention networks are engaged.

3. Hippocampal Involvement in Memory: The significant involvement of the hippocampus in memory formation is underscored by Supplementary Tables 1, 5, 8, and 9. However, the paper lacks a discussion on how attention may influence hippocampal activation. Addressing this could resolve outstanding questions and enhance the paper's contribution to the literature on memory systems.

We agree with the reviewer that an involvement of the hippocampus was, in principle, expected by the design of the memory task and its previously described role in the subsequent memory effect (Kim, 2011). However, the current study did not find significant hippocampal hemodynamic responses in the post-error subsequent memory effect. The absence of an effect in the hippocampus in this analysis cautioned us to refrain from speculations on attention-dependent variations of hippocampal involvement. Since previous studies have associated the hippocampus with reconsolidation processes, we performed an additional analysis between repeated correct and initial correct trials, to investigate which regions may be related to the stabilization of already correct memory representations. In this analysis, the left hippocampus showed a large-spread effect spanning along the anterior-posterior axis, showing an involvement in reconsolidation but not in error-driven learning success. We included an additional supplementary figure (see Supplementary figure 3) showing the univariate general linear model results and included a supplementary cluster table (see Supplementary Table 16).

Methods (p. 10 l. 248): To investigate neurophysiological associations of reconsolidation processes and error-driven learning successes, an additional analysis assessed differences between subsequent repeated correct and initial correct trials ($\text{Correct}_{\text{HighConfidence}} - \text{Correct}_{\text{HighConfidence}} > \text{Error}_{\text{LowConfidence}} - \text{Correct}_{\text{HighConfidence}}$) and the results are included in the supplementary materials (Supplementary figure 3 and Supplementary table 16).

Results (p. 17, l. 417): Surprisingly, hippocampal effects were not found in the post-error SME, although we found an involvement of the hippocampus in additional analyses on memory reconsolidation [$\text{Correct}_{\text{HighConfidence}} - \text{Correct}_{\text{HighConfidence}} > \text{Error}_{\text{LowConfidence}} - \text{Correct}_{\text{HighConfidence}}$, see Supplementary Figure 3 and Supplementary Table 17].

Supplementary Figure 3 | General linear model results differentiating between $\text{Correct}_{\text{ConfidenceHigh}}$ and $\text{Error}_{\text{ConfidenceLow}}$ trials. Increased hemodynamic responses (red) are found in the hippocampus and medial frontoparietal default mode network regions, suggesting neurophysiological underpinnings of reward-based reconsolidation processes. Decreased hemodynamic responses (blue) are found in regions of the midcinguloinsular salience/ventral attention network nodes such as anterior insula and posterior medial frontal cortex, suggesting regions involved in initial error-driven association learning success.

4. Task Design and Memory Encoding: The study focuses on an associative memory task linking faces with Gabor patches, yet the discussion primarily addresses the encoding of faces, neglecting the associative aspect of the task. This oversight could mislead interpretations of the task's requirements and its impact on memory performance. A more balanced discussion of both components—face and association—would align better with the task's design and objectives.

We agree with the reviewer that the analysis focused on stimulus-processing evidence of the memory-relevant cue category (faces), while respective analysis for the target stimuli (tilted gabor patches) were not performed. We added a paragraph in the limitation section to elaborate on why the study design was not suitable to address the level of gabor patch orientation evidence in the same manner.

Limitations (p. 28 l. 676): The current study investigated neurophysiological associations of memory-related demand detection processes, associative learning improvements and memory-relevant stimulus processing evidence. Because of the abundant literature on ventral visual stream regions showing face-specific

processing, such as in FFA, the study was designed to assess face-processing evidence during associative memory formation based on relevant cytoarchitectonic masks of the fusiform gyrus. In this regard, face stimuli were the presented memory cues which were to be associated with different target orientations of gabor patches. While the study showed an error-driven and subsequent recall success-related upregulation of face processing in the fusiform gyrus, respective analyses on the target orientations of gabor patches were not performed. Although previous studies were able to decode orientations especially from visual cortical regions, the low number of repetitions per particular orientation of respective gabor patches ($n = 7$) prevented us from fitting robust machine learning models. Therefore, it cannot be concluded that the same regions upregulate the processing of association-memory cue and target stimuli. Further studies are needed to assess the generalizability and robustness of memory-relevant cognitive control regions and their replicability for different stimulus categories.

5. Statistical Dependence Across Epochs: The paper examines five types of epochs—Recall, Confidence, Feedback, Encoding, and ITI—which correspond to distinct phases of the trials. The potential for statistical dependence among these epochs warrants discussion. It would be beneficial for the authors to explain how they managed this in their experimental design and to acknowledge any limitations this might impose on the study's findings.

We agree with the reviewer about the challenge to distinguish different cognitive processes within quickly overlapping events when measured by fMRI and based on blood-oxygen-level-dependent effects. We clarified our strategy on assessing statistical independence in the methods section and conducted further analyses on the design matrix regressors intercorrelations, which were provided as additional figures in the supplementary materials. We included another paragraph in the limitation section to caution against interpreting the specificity for, e.g., neurophysiological underpinnings of different error evidence sources. We also referenced to other studies with more targeted designs for this purpose.

a

b

Supplementary Figure 1 | Correlation matrix of the design matrix regressors. The matrix shows the average Pearson correlation between different trial types in the GLM for a, error-monitoring processes and b, post-error subsequent memory effect (SME).

Methods (p. 10 l. 228): In the first GLM, neurophysiological signals related to recall, confidence and feedback were included and each trial type was convolved as a separate regressor, such that the shared variance is encompassed in the residual variance of the model. Multicollinearity between convolved regressors was examined using the variance-inflation-factor index, assuming moderate multicollinearity for

values > 5 and < 10 , and high multicollinearity for a variance-inflation-factor > 10 , and using Pearson correlation values below $< .90$ following current practices to indicate sufficiently efficient design matrices^{32, 33}.

(p. 10 l. 241): In the second GLM, encoding regressors were used together with regressors for recall and for confidence while the feedback-related regressors were excluded because of the redundancy and temporal overlap with encoding regressors.

(p. 12 l. 289): First, the univariate GLMs described in the previous sections were adapted for single-trial deconvolution according to the least-squares separate approach³⁶ to obtain a series of beta-maps. In this regard, all correct rejection face and house trials in the localizer task were determined and stimulus presentation of each trial was once defined as target event in an additional, independent GLM. The target trial was convolved with a hemodynamic response function as a separate regressor, while controlling for all other events and denoising parameters such as in univariate GLM analyses.

(p. 13 l. 324): Single-trial deconvolution and selection of FFA voxels was repeated for the 160 trials in FALT and the three memory-relevant epochs of stimulus recall, encoding and ITI. To ensure sufficient independence and additional variance explanation of the epochs, a step-wise procedure was chosen, such that in the recall-related deconvolution only recall events were included, in the encoding-related deconvolution both recall and encoding events for included and in the ITI-related deconvolution all three events were included.

Results (p.17 l. 401): Variance-inflation-factor indices were < 5 and $R_{\text{Pearson}} < .90$ between the convolved design matrix regressors, indicating sufficiently low multicollinearity in the univariate GLM analysis (Supplementary Figure 1).

Limitations (p. 28 l. 697): The results of this study showed that the post-error SME contrast was overlapping with a posterior, superior portion of pMFC as found in the contrast related to negative feedback processing, but not to the exact location of the error-monitoring conjunction per se. Overall, pMFC overlaps with the processing of negative feedback were abundant also in relation to increased face processing evidence. While negative feedback represents the most explicit evidence for required learning improvements and adaptations during the following encoding epoch, it was not the primary purpose of the study to differentiate sources of memory-error evidence accumulation processes, such as internal evidence from confidence levels and external evidence from negative feedback. Control analyses suggested sufficiently low levels of multicollinearity to assess respective contrasts and different epochs, potentially based on self-paced motor responses in the selection periods during task performance which separate different cognitive events within the same trials. However, adapted task designs will be better suited to investigate how different origins of memory-error evidence differ in their hemodynamic topography. Future studies addressing this research question may benefit from assessing inter-individual variations in paracingulate gyrification patterns to reveal under which circumstances which pMFC subregions are associated with particular performance monitoring processes, jointly or specifically.

6. Classification Rates Explanation: On page 13, the reported classification rates for face presentation during different epochs are notably low (39.53% during encoding and 17.35% during rehearsal). These figures require

clarification regarding their implications. Are these rates considered successful, or do they indicate limitations in the classifier's performance? An explanation would aid in understanding the effectiveness of the employed methods.

We agree with the reviewer about the difficulty to interpret the number of predicted face trials as an accuracy score. We added a paragraph in the limitation section to discuss potential challenges in the cross-classification analysis based on the used cognitive paradigms, and elaborate on the verification of additional quality criteria to ensure that the studies implications are reasonably derived from the results and analytical strategy.

Limitations (p.29 l. 716): Previous studies conducting multivariate fMRI analyses have often used classification accuracies based on a binary classification on a given trial instead of more fine-grained parametric measure such as class probabilities²² or measures such as the distance from a multivariate hyperplane. Here, we applied multivariate cross-classification analyses to investigate whether and how the level of face-processing evidence in an associative learning task is varying in relation to detected task demands and successful memory formation adaptations as substantiated by a correct subsequent recall. While the localizer task and respective training of the multivariate models contained the same number face and house stimuli, in FALT only face stimuli were used as memory cues. Accordingly, classification accuracies in the FALT should be interpreted as the relative face-processing evidence and classification rates below 50% do not reflect chance level performance. Validity of the model was confirmed by robust cross-validation with balanced accuracies in the localizer task, a systematic inter-individual link between cross-classification prediction rates and the average scaled face-class probability, as well as replicable hemodynamic topographies associated with face processing evidence during different memory-relevant epochs. Most importantly, external validity of the model's scaled probability function was indicated by the separation of behavioral relevance measures such as encoding necessity and subsequent recall success. Overall, quantifying the relative level of evidence face-processing instead of applying discretized binary classification accuracies enabled to show intra-individual variations in stimulus evidence on a single-trial level and can, therefore, provide a link to memory-relevant attention allocation and cognitive control processes.

Supplementary Figure 2 | Relationship between the average rate of face predictions per participants and average single-trial face-class probability. To validate the cross-classification-based single-trial stimulus evidence parameter, inter-individual

associations were assessed. Cross-classification prediction rates and average class probability showed a strong association (Spearman correlation > .90) during all three epochs, a, recall, b, encoding and c, inter-trial-interval (ITI).

7. Clarification of Face-processing Evidence Statistics: On page 16, the reported increases in face-processing evidence (3.2%) and recall success (1.1%) derived from multivariate classification evidence need clarification. Are these increases indicative of improvements in classification accuracy? Detailing this would help in interpreting the statistical significance and practical implications of these findings.

We thank the reviewer for addressing the need for clarification on the statistical results, where factors such as encoding demand and subsequent recall success have been fit to the face-processing evidence measure. We added a more detailed explanation to clarify the meaning of the reported percentage values both in the results and methods sections, and included the Fig. 5b and the linear mixed model table regarding encoding epochs as an example for the sake of completeness. For more details on assessing the validity of the measure and interpretations of interpreting the prediction's variability, please see our response to the previous comment.

Fig. 5 | (...) b, The level of evidence for face-processing was higher when there was a demand of memory improvement during recall and encoding, and significantly higher for subsequent recall success during encoding epochs, as found in the linear mixed model results.

Supplementary table 11 | Encoding-related face representation strength. Mixed linear model regression results fit to the probability-scaled likelihood of face-processing during deconvolved single-trial encoding epochs.

Model:	MixedLM	Dependent variable	Face Probability (Encoding)			
Number of observations:	3183	Method:	REML			
Number of groups:	28	Scale:	0.0654			
Minimal group size:	101	Log-Likelihood:	-235.6007			
Maximal group size:	120	Converged:	Yes			
Mean group size:	113.7					
	Coefficient	Standard error	z	p	[0.025	0.975]
Intercept	0.435	0.026	16.423	<0.001	0.383	0.487
Encoding demand	0.039	0.005	7.176	<0.001	0.028	0.049
Subsequent recall success	0.015	0.007	2.254	0.024	0.002	0.028
Group variable	0.019	0.021	-	-	-	-

Methods (p. 14 l. 350): In particular, the encoding demand regressor was set to a value of 0 for CorrectCorrect trials, and to a value of 1 for ErrorError and ErrorCorrect trials. The regressor for subsequent recall success was set to a value of 0 for ErrorError trials, and to a value of 1 for ErrorCorrect and CorrectCorrect trials. The face-class processing measure was based on a Platt-scaled probability measure of the linear support vector machine, and was able range between 0 and 1, where 1 indicates the highest level of multivariate evidence for FFA-based face-processing. In this way, beta-weights from results from the linear mixed model corresponded to values interpretable as increased face-processing evidence according to a change in encoding demand and in association with subsequent recall success.

Results (p. 22 l. 543): The percentage values in this case mean that for example within encoding epochs, there is a 3.9 % stronger level of face-processing evidence in trials which require improved encoding, such as ErrorError and ErrorCorrect trials, compared to CorrectCorrect trials. On the other hand, face-processing evidence is 1.5 % higher on those trials of a participant which are the learning opportunities for subsequent successful recall, i.e., ErrorCorrect and CorrectCorrect trials compared to ErrorError trials.

References

Kim, H., *Neural activity that predicts subsequent memory and forgetting: a meta-analysis of 74 fMRI studies.* *Neuroimage*, 2011. 54(3): p. 2446-61.

Mumford, J.A., J.-B. Poline, and R.A. Poldrack, *Orthogonalization of Regressors in fMRI Models.* *PLOS ONE*, 2015. 10(4): p. e0126255.

Uddin, L.Q., B.T.T. Yeo, and R.N. Spreng, *Towards a Universal Taxonomy of Macro-scale Functional Human Brain Networks.* *Brain topography*, 2019. 32(6): p. 926-942.

Uddin, L.Q., et al., *Controversies and progress on standardization of large-scale brain network nomenclature.* *Network Neuroscience*, 2023. 7(3): p. 864-905.

Walther, A., et al., *Reliability of dissimilarity measures for multi-voxel pattern analysis.* *Neuroimage*, 2016. 137: p. 188-200.

Yeo, B.T., et al., *The organization of the human cerebral cortex estimated by intrinsic functional connectivity.* *J Neurophysiol*, 2011. 106(3): p. 1125-65.